# FLIP: Benchmark tasks in fitness landscape inference for proteins

**Christian Dallago**[*]
Technical University of Munich
christian.dallago@tum.de

**Jody Mou**[*]
Microsoft Research New England
jodymou@mit.edu

**Kadina E. Johnston**
BBE, Caltech
kjohnston@caltech.edu

**Bruce J. Wittmann**
BBE, Caltech
bwittman@caltech.edu

**Nicholas Bhattacharya**
UC Berkeley
nick_bhat@berkeley.edu

**Samuel Goldman**
CSB, MIT
samlg@mit.edu

**Ali Madani**
Saleforce Research
amadani@salesforce.com

**Kevin K. Yang**
Microsoft Research New England
yang.kevin@microsoft.com

## Abstract

Machine learning could enable an unprecedented level of control in protein engineering for therapeutic and industrial applications. Critical to its use in designing proteins with desired properties, machine learning models must capture the protein sequence-function relationship, often termed *fitness landscape*. Existing benchmarks like CASP or CAFA assess structure and function predictions of proteins, respectively, yet they do not target metrics relevant for protein engineering. In this work, we introduce Fitness Landscape Inference for Proteins (FLIP), a benchmark for function prediction to encourage rapid scoring of representation learning for protein engineering. Our curated tasks, baselines, and metrics probe model generalization in settings relevant for protein engineering, e.g. low-resource and extrapolative. Currently, FLIP encompasses experimental data across adeno-associated virus stability for gene therapy, protein domain B1 stability and immunoglobulin binding, and thermostability from multiple protein families. In order to enable ease of use and future expansion to new tasks, all data are presented in a standard format. FLIP scripts and data are freely accessible at https://benchmark.protein.properties.

## 1 Introduction

Proteins are life's workhorses, efficiently and precisely performing complex tasks under a wide variety of conditions. This combination of versatility and selectivity makes them not only critical to life, but also to a myriad of human-designed applications. Engineered proteins play increasingly essential roles in industries and applications spanning pharmaceuticals, agriculture, specialty chemicals, and fuel [1–5]. The ability of a protein to perform a desired function is determined by its amino acid sequence, often mediated through folding to a three-dimensional structure [6]. Unfortunately, current biophysical and structural prediction methods cannot reliably map a sequence to its ability to perform a desired function, termed protein *fitness*, with sufficient precision to distinguish between closely-related protein sequences performing complex functions such as catalysis. Therefore, protein engineering has relied heavily on directed evolution (DE) methods, which stochastically modify ("mutate") a starting sequence to create a library of sequence variants, measure all variants to find those with improved fitness, and then iterate until the protein is sufficiently optimized [7]. Directed evolution is energy-, time-, and material-intensive, in part because it discards information from

---

[*] Equal contribution

unimproved sequences. Machine-learning methods that predict fitness from sequence can leverage both positive and negative data to intelligently select variants for screening, reaching higher fitness levels with fewer measurements than traditional directed evolution, and without necessarily requiring detailed understanding of structure or mechanism [8, 7, 9–11].

Directed evolution campaigns are often limited by the cost of collecting sequence-fitness data. Therefore, machine learning approaches for sequence-fitness prediction are most useful in protein engineering when they can learn from low-N (few sample) labeled datasets or when they can generalize to types of variation that are unobserved in the training set. Rapid advances in genomic sequencing technology have led to an explosion of putative protein sequences [12, 13] deposited in databases like UniProt [14]. Recent efforts in sequence-function prediction [15, 16] have sought to leverage the information in these unlabeled sequences through pretraining and fine-tuning, and have successfully engineered proteins with brighter fluorescence and high catalytic efficiency [17]. Unsupervised models were also applied to- or built on evolutionary sequence inputs to model the effects of mutations [18–21].

In this work, we present a suite of benchmarking tasks for protein sequence-fitness prediction with the dual aims of enabling protein engineers to compare and choose machine learning methods representing protein sequences and accelerating research on machine learning for protein fitness prediction. Our tasks are curated to be diverse in the functions measured and in the types of underlying sequence variation. For each landscape, we provide one or more train/test splits that evaluate biologically-relevant generalization and mimic challenges often seen in protein engineering. Figure 1 and Table 2 summarize the landscape tasks and splits. We also compute the performance of baseline models against which future models can be compared, and which highlight that our tasks can distinguish between "better" and "worse" pretraining regimes. Landscapes and baselines are available at https://benchmark.protein.properties, while a glossary technical terms is provided in the supplement.

## 2   Related Work

Well-designed and easily accessible benchmarks have encouraged and measured progress in machine learning on proteins, especially protein structure prediction. The Critical Assessment of Protein Structure Prediction (CASP) [22], and retrospective protein training datasets from previous CASP competitions [23] have lowered the barrier to entry for new research teams and provided a clear account of progress over the last three decades [24]. DeepMind's recent landmark results with their *AlphaFold2* predictor in CASP 14 [25] built on these community-driven efforts.

Table 1: Performance (Spearman's correlation) on TAPE engineering tasks. Performances reported in referenced literature. CNNs were replicated from [26] without test set clipping.

|  | Pretraining | Fluorescence | Stability |
| --- | --- | --- | --- |
| ESM [27] | masked language model | 0.68 | 0.71 |
| TAPE transformer [28] | masked language model | 0.68 | 0.73 |
| TAPE LSTM [28] | bidirectional language model | 0.67 | 0.69 |
| TAPE ResNet [28] | masked language model | 0.21 | 0.73 |
| UniRep [29] | language model + structure | 0.67 | 0.73 |
| CPCProt [30] | contrastive | 0.68 | 0.65 |
| CPCProt-LSTM [30] | contrastive | 0.68 | 0.68 |
| Linear regression [26] | none | 0.68 | 0.48 |
| CNN [26] | none | 0.67 | 0.51 |
| Mutation count [31] | none | 0.45 | NA |
| BLOSUM62 score [31] | none | 0.50 | NA |

Inspired by the effectiveness of CASP, there have been attempts at benchmarks for function prediction and protein pretraining. The Critical Assessment of Function Annotation (CAFA) [32, 33] focuses on assigning Gene Ontology (GO) classes (categorical definitions of protein functions) to proteins. While an important benchmark, CAFA does not directly require models to build on sequence inputs, instead

they could leverage graph inputs from protein-protein interaction networks, and the prediction targets do not account for fitness variations between very similar sequences that are important for protein engineering. Tasks Assessing Protein Embeddings (TAPE) [28] aims to evaluate the effectiveness of different pretraining regimes and models to predict protein properties. Of the five tasks in TAPE, three (remote homology, secondary structure, and contacts) focus on structure prediction, while only two (fluorescence and stability) target fitness prediction. These two tasks show little discriminative power between different models [26], as shown in Table 1. In addition, the use of structure as an evaluation limits the creation of jointly trained structure- and sequence- based embeddings that may be most useful in protein engineering tasks [34]. Envision [35] collates several dozen single amino-acid variation (SAV) datasets, but does not include other types of sequence variation of interest to protein engineers. DeepSequence [19] collects 42 deep mutational scan (DMS) datasets for evaluation purposes. These capture single and multiple co-occurring residue substitutions, but do not capture variation at the proteome scale, or mutational paths from large insertions and deletions. Furthermore, while DMS landscapes may characterize the effect of co-occurring substitutions, not every sample with co-occurring residue substitutions may express these at sites relevant for a measured function, and in turn, evaluations on all possible co-occurring substitutions may not always be expressive (e.g., if the measured function is binding and a sample has two substitutions, one at a residue at the interface and one elsewhere, the effect may still be high simply because an interface residue is involved). Finally, the data from these studies does not come with standard column headers or train/test splits, hindering use in automated evaluation pipelines.

The limitations of the existing benchmarks have led pretraining methods to be primarily evaluated by their ability to predict structural information [36, 37]. While the ability to impart structural knowledge through sequence-only pretraining is impressive, it is not the most important criterion for protein engineers. Efforts to systematically compare new methods on fitness prediction have required researchers to both gather their own collection of datasets and compute their own baseline comparisons [16, 38–40].

## 3  Landscapes and Splits

We design FLIP to answer two fundamental questions about machine learning model learning protein sequences:

1. Can a model capture complex fitness landscapes beyond mutations to a parent sequence?

2. Can a model perform well across a range of proteins where fitness is measured for very different functions?

Existing work such as DeepSequence [19] and Envision [35] succeed at the second criterion but not the first. TAPE [28], on the other hand, evaluates the first criterion with its fluorescence task but not the second. We prioritized complex landscapes (with insertions and deletions) rather than single amino acid variants (e.g. deep mutational scans), to practically cover a larger sequence space, as well as potentially more functional diversity finalized to ensure model generalization for broad applicability.

To test the aforementioned questions, we collect three published landscapes and create 15 corresponding dataset splits as desribed in the following and summarized in Table 2. We choose landscapes and splits that cover a broad range of protein families, sequence variation, and fitness landscapes with rigorous measurements. Each landscape is transformed into one or more splits to test different model generalization abilities, as shown in Figure 1; many of the splits were also made to reflect standard laboratory data-collection practices, thus testing the appropriateness of models to real-world applications.

Simple random splits are notoriously misleading in classical protein sequence-to-function prediction as protein sequences are not sampled I.I.D., but with correlations induced by evolutionary history. This means that random splits reflect a notion of generalization not of interest to most biologists [46]. While there are standard heuristics for approximating the correlation structure due to evolution (such as sequence-identity deduplication\redundancy reduction), in the protein engineering setting there are not similarly standardized approaches. As such, we resorted to landscape-specific approaches informed by the conditions of each experiment, as detailed in Figure 1.

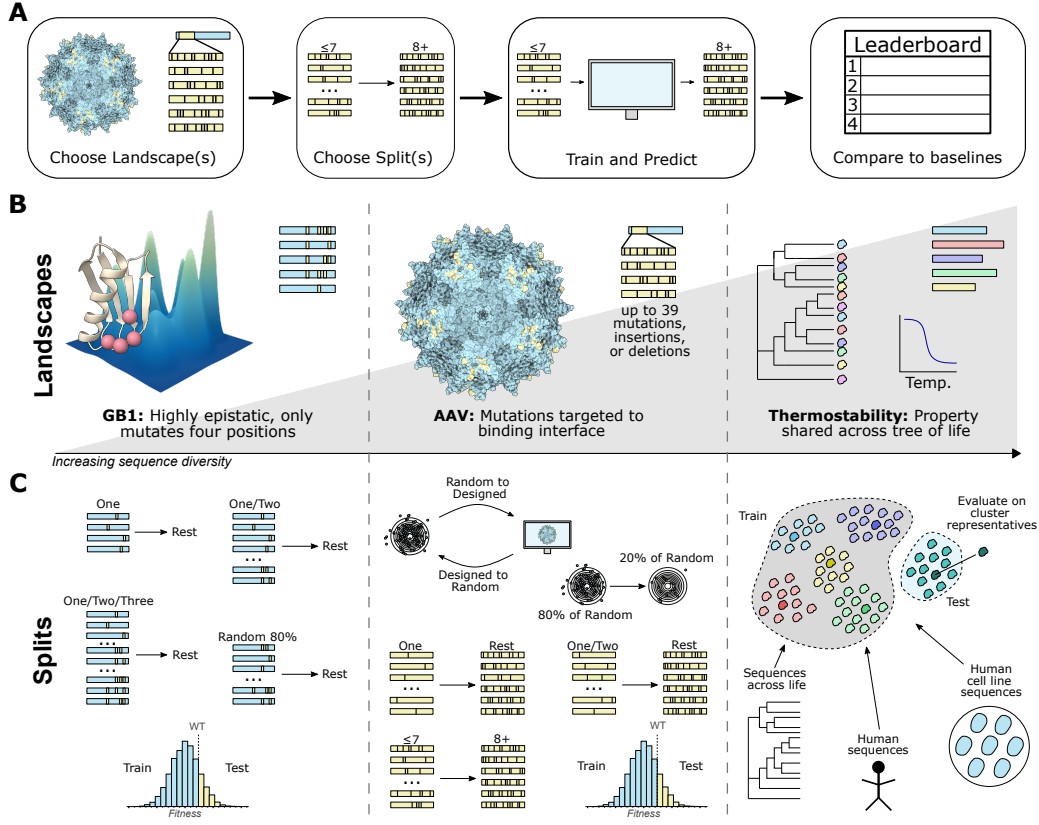

Figure 1: Summary of the workflow, landscapes, and splits. (A) General FLIP workflow: choose landscapes and splits that match user needs, train models and make predictions on the test set, and then compare to baseline models. (B) We choose landscapes that cover different types of sequence diversity. The GB1 landscape focuses on simultaneous mutation of four epistatic sites with nearly complete coverage [41] (PDB ID: 2GI9 [42]). The AAV capsid protein landscape sparsely samples sequences with up to 28 mutations, including insertions and deletions, to the the binding interface [43] (PDB ID: 6IH9 [44]). The thermostability landscape [45] measures a property shared by proteins from multiple functional groups across different domains of life. (C). We also provide up to seven suggested data splits for each landscape, which are described in Section 3.

Table 2: Landscapes and split statistics. The sampled splits (*) are mainly used for discourse in this manuscript, as such splits are rarely observed in practice when working with biological data.

| Landscape | Split | Total samples | Train samples | Test samples |
|---|---|---|---|---|
| AAV | Mut-Des | 284,009 | 82,583 | 201,426 |
| | Des-Mut | 284,009 | 201,426 | 82,583 |
| | 1-vs-rest | 82,583 | 1,170 | 81,413 |
| | 2-vs-rest | 82,583 | 31,807 | 50,776 |
| | 7-vs-rest | 82,583 | 70,002 | 12,581 |
| | low-vs-high | 82,583 | 47,546 | 35,037 |
| | Sampled* | 82,583 | 66,066 | 16,517 |
| Thermostability | Mixed | 27,951 | 24,817 | 3,134 |
| | Human | 10,093 | 8,148 | 1,945 |
| | Human-cell | 7,156 | 5,792 | 1,366 |
| GB1 | 1-vs-rest | 8,733 | 29 | 8,704 |
| | 2-vs-rest | 8,733 | 427 | 8,306 |
| | 3-vs-rest | 8,733 | 2,968 | 5,765 |
| | low-vs-high | 8,733 | 5,089 | 3,644 |
| | Sampled* | 8,733 | 6,961 | 1,772 |

The vast majority of representation learning on protein sequences models entire sequences [27, 37, 15, 34]. As such, we use entire protein sequences as inputs, even for landscapes derived from studies examining mutations at a small subset of positions. While we include a naïve validation set for each split for comparison purposes, we encourage users to engineer their own validation splits from the training data. All tasks and splits are provided in a consistent, easy-to-use CSV format and are available at https://benchmark.protein.properties. Original datasets were either supplemented to published research (Wu et al.) under CC BY 4.0, or were obtained with written permission from the authors (Jarzab et al., Bryant et al.). Data derivatives proposed as tasks are licensed under AFL-3.

## 3.1 GB1

**Motivation.** One challenge confronting protein engineering is the ability to predict the effects of interactions between mutations, termed epistasis. These interactions result in non-additive effects on protein fitness and have been shown to constrain the paths available to evolution, especially evolution via a greedy walk. Furthermore, as more mutations are made simultaneously, these interactions become more complex and more difficult to predict. Therefore, we wish to assess model predictions on an exhaustive, combinatorial, and highly epistatic mutational landscape, focusing on learning from variants with fewer mutations to predict the activity of variants with more mutations.

**Landscape.** We use the GB1 landscape [41], which has become a gold standard for investigating epistatic interactions [10]. GB1 is the binding domain of protein G, an immunoglobulin binding protein found in Streptococcal bacteria [47, 48]. In their original study, Wu et al. measured the fitness of $149,361$ of $160,000$ possible combinations of mutations at 4 positions.

**Splits.** Over $96\%$ of the amino acid mutations in this set yield non- or poorly-binding sequences – $143,539$ out of $149,361$ sequences have fitness value below $0.5$, where wild-type fitness is 1 and a fitness of $0$ is non-binding. Thus, models trained on the full experimental data can achieve high performance by predicting low fitness regardless of inputs. To ensure that models learn nontrivial signal, we downsample non-functional sequences prior to creating the training sets. Specifically, we include all $5822$ sequences with fitness above $0.5$ and $2911$ randomly-sampled sequences with fitness less than or equal to $0.5$. From this set, we curate five dataset splits to test generalization from few-mutation sequences to many-mutation sequences, from low fitness to high, and one extra randomly sampled split for discussion purposes:

- **Train on single mutants (1-vs-rest):** Wild type and single mutants are assigned to train, while the rest are assigned to test. This split is one of the most commonly observed in an applications setting, where a researcher has gathered data for many single mutations of interest and wishes to predict the best combinations of mutations.

- **Train on single and double mutants (2-vs-rest):** Wild type, single and double mutants are assigned to train, while the rest are assigned to test. This is also a commonly observed split in an applications setting, albeit, at a lesser frequency than 1-vs-rest.

- **Train on single, double and triple mutants (3-vs-rest):** Wild type, single, double and triple mutants are assigned to train, while the rest are assigned to test.

- **Train on low fitness, test on high (low-vs-high):** Sequences with fitness value equal or below wild type are used to train, while sequences with fitness value above wild type are used to test.

- **Sampled:** Sequences are randomly partitioned in 80% train and 20% test. This split serves mostly for discussion purposes in this manuscript.

## 3.2 AAV

**Motivation.** Mutations for engineering are often focused in a specific region of a protein. For example, this is done if a protein-protein interface is known to be at a subset of positions. Successfully predicting fitness for a long sequence being mutated at a subset of positions is a task of wide applicability.

**Landscape.**  Adeno-associated virus (AAV) capsid proteins are responsible for helping the virus integrate a DNA payload into a target cell [49], and there is great interest in engineering versions of these proteins for gene therapy [43, 50, 51]. Bryant et al. prepared a rich mutational screening landscape of different VP-1 AAV proteins (UniProt [14] Accession: P03135), and this data has been successfully used as a basis for machine learning-guided design [52, 53]. In their study, Bryant et al. mutagenize a 28-amino acid window from position 561 to 588 of VP-1 and measure the fitness of resulting variants with between 1 and 39 mutations, which we refer to as the *sampled* pool. In addition they measured the fitness of sequences chosen or designed using various machine-learning models. We refer to these as the *designed* pool.

**Splits.**  We derive seven splits from this landscape that probe model generalization:

- **Sampled-designed (Mut-Des):** All *sampled* sequences are assigned to train; all *designed* sequences are assigned to test.

- **Designed-sampled (Des-Mut):** All *designed* sequences are assigned to train; all *sampled* sequences are assigned to test.

- **Train on single mutants (1-vs-rest):** Wild type and single mutants in the *sampled* pool are assigned to train, while the rest are assigned to test. As with the GB1 1-vs-rest split, this reflects a common dataset split observed in protein engineering applications.

- **Train on single and double mutants (2-vs-rest):** Wild type, single and double mutants in the *sampled* pool are assigned to train, while the rest are assigned to test. Again, as with the GB1 2-vs-rest split, this reflects a common dataset split observed in protein engineering applications.

- **Train on mutants with up to seven changes (7-vs-rest):** Mutants with up to and including seven changes in the *sampled* pool are assigned to train, while the rest are assigned to test.

- **Train on low fitness, test on high (low-vs-high):** For sequences in the in the *sampled* pool, sequences with fitness value equal or below wild type are used to train, while sequences with fitness value above wild type are used to test.

- **Sampled:** Sequences in the *sampled* pool are randomly partitioned in 80% train and 20% test. This split serves mostly for discussion purposes in this manuscript.

### 3.3  Thermostability

**Motivation.**  Thermostability is very often a desirable trait that complements more application-specific functions. For example, thermostable enzymes not only allow operation at higher reaction temperatures with faster reaction rates, but are also better starting points for directed evolution campaigns [54, 55]. This explains why thermostability has been a consistent target for multi-objective optimization in protein engineering [56–58]. Thermostability can be challenging to predict, because it is not necessarily a smooth function landscape; in certain protein families, a single amino acid substitution can confer or destroy thermostability [59].

**Landscape.**  We curate an extensive screening landscape from the Meltome Atlas [45], which used a mass spectrometry-based assay to measure protein melting curves across 13 species and 48,000 proteins. Unlike the other landscapes, which measure the effects of sequence variation from a single starting point on a function specific to that protein, this landscape includes both global and local variation.

**Splits.**  We derive three splits from this landscape, considering biological realities and common dataset regularizations for cross-spices and sequence-diverse sets:

- **Mixed:** We cluster all available sequences and select cluster representatives using MM-seqs2 [12] at a threshold of 20% sequence identity to create one split. In this split, all sequences in 80% of clusters are assigned to train, while only cluster representatives from the remaining 20% of clusters are assigned to test.

- **Human:** We cluster sequences in human and select cluster representatives using MM-seqs2 [12] at a threshold of 20% sequence identity to create one split. In this split, all

sequences in 80% of clusters are assigned to train, while only cluster representatives from the remaining 20% of clusters are assigned to test.

- **Human-cell:** We cluster sequences of one cell line for human and select cluster representatives using MMseqs2 [12] at a threshold of 20% sequence identity to create one split. In this split, all sequences in 80% of clusters are assigned to train, while only cluster representatives from the remaining 20% of clusters are assigned to test.

## 4    Baseline algorithms

We evaluate three major groups of baselines (Table 3) – parameter-free, supervised, and pretrained. These three classes correspond to common approaches from different communities. In particular, we seek to clarify the value of transfer learning for protein engineering by benchmarking pretrained models against purely supervised methods systematically. We also hope to simplify algorithm selection for practitioners by providing a single place to compare many commonly used methods. Note that we do not use Potts models [60], popular in protein structure prediction [61], because of the need to build high-quality multiple sequence alignments, which would be impractical for the thermostability dataset. Furthermore, Potts models use artificial constructs when dealing with datasets with large insertions and deletions (e.g., modeling sequence deletions through special characters), as is the case for the AAV landscape. However, in the presence of well curated MSAs, these approaches can be successful in modeling the effect of residue substitutions [62].

Table 3: Baseline methods

| Method | Description |
| --- | --- |
| Levenshtein | Levenshtein distance to wild-type. |
| BLOSUM62 | BLOSUM62-score relative to wild-type. |
| Ridge regression | Ridge regression model on one-hot encoding. |
| Convolutional network | Simple convolutional network on one-hot encoding. |
| ESM-untrained | 750M parameter transformer with randomly-initialized weights |
| ESM-1b [27] | 750M parameter transformer pretrained on UniRef50. |
| ESM-1v [16] | 750M parameter transformer pretrained on UniRef90. Only **one** element of ensemble used due to compute constraints. |

For baselines using protein language models, which compute an embedding for every amino acid, we pool embeddings in three ways:

- **Per amino acid (per AA):** A supervised model is tasked to learn how to pool over the sequence using a 1D attention layer to return a regression prediction.

- **Mean:** Sequence embeddings are mean pooled per amino acid over the length of the protein sequence to obtain a fixed-size input for each sequence.

- **Mean over subset (mut mean):** Sequence embeddings are mean pooled per amino acid for the residues in the mutated region of interest to obtain a fixed-size, region specific input from the sequence.

To train the models, 10% of each training set is sampled at random as a validation set. For *Ridge*, we use the scikit-learn implementation of ridge regression with default parameters. The CNN consists of a convolution with kernel width 5 and 1024 channels, a ReLU non-linearity, a linear mapping to 2048 dimensions, max pool over the sequence, and a linear mapping to 1 dimension. CNNs are optimized using Adam [63] with a batch size of 256 (GB1, AAV) or 32 (thermostability) and a learning rate of 0.001 for the convolution weights, 0.00005 for the first linear mapping, and 0.000005 for the second linear mapping. Both linear mappings have a weight decay of 0.05. For ESM models, by far the most computationally expensive baselines, we train with a batch size of 256, a learning rate of 0.001, and the Adam optimizer. CNNs and the ESM models are trained with early stopping with a patience of 20 epochs. Models are trained on a NVidia Quadro RTXA6000 GPU. Code, data, and instructions needed to reproduce results can be found at https://benchmark.protein.properties.

# 5 Results

Overall, we observe that for landscapes around a wild type (Tables 4 & 5), pretraining offered by ESM-1b [27] or ESM-1v [16] does not help much when sufficient training data is available (see Table 2 for statistics), at least in the setting explored here: using these protein language models to collect *frozen* embeddings as inputs to subsequent prediction models. Conversely, for the split involving diverse sequences (Table 6), pretraining yields a large boost over pure supervision. The best method of pooling residue-embeddings for whole sequences varies depending on task (Table 4, 5 & 6). Most remarkably, training simple models (CNN, ridge regression) is competitive over a wide range of regimes. We exclude results for *per-AA* ESM models for the AAV *Des-Mut* task (Table 5), as we estimated that it would require a month of compute using for Nvidia A6000 GPUs, which appeared unjustified for a baseline metric computation. Hyperparameter search results are reported in the supplement, as are evaluations using different metrics.

Table 4: GB1 baselines (metric: Spearman correlation)

| Model | 1-vs-rest | 2-vs-rest | 3-vs-rest | low-vs-high |
|---|---|---|---|---|
| ESM-1b (per AA) | 0.28 | 0.55 | 0.79 | 0.59 |
| ESM-1b (mean) | 0.32 | 0.36 | 0.54 | 0.13 |
| ESM-1b (mut mean) | -0.08 | 0.19 | 0.49 | 0.45 |
| ESM-1v (per AA) | 0.28 | 0.28 | 0.82 | 0.51 |
| ESM-1v (mean) | 0.32 | 0.32 | 0.77 | 0.10 |
| ESM-1v (mut mean) | 0.19 | 0.19 | 0.80 | 0.49 |
| ESM-untrained (per AA) | 0.06 | 0.06 | 0.48 | 0.23 |
| ESM-untrained (mean) | 0.05 | 0.05 | 0.46 | 0.10 |
| ESM-untrained (mut mean) | 0.21 | 0.21 | 0.57 | 0.13 |
| Ridge | 0.28 | 0.59 | 0.76 | 0.34 |
| CNN | 0.17 | 0.32 | 0.83 | 0.51 |
| Levenshtein | 0.17 | 0.16 | -0.04 | -0.10 |
| BLOSUM62 | 0.15 | 0.14 | 0.01 | -0.13 |

**GB1.** Table 4 summarizes baseline results for the biologically motivated GB1 splits. When models are trained only on single mutants, all variations on ESM-1b [27] and ESM-1v [16] outperform supervised models. This regime has little training data (29 samples, Table 2), giving the most opportunity for pretraining to compensate. The difference between pretrained and supervised models largely disappears once models are trained on both single and double mutants (2-vs-rest, Table 4). The various pooling choices for embeddings perform inconsistently across datasets and splits; for example, *mut-mean* does best on 1-vs-rest but worst on 3-vs-rest. The *low-vs-high* split suggests. The sampled split reported separately in Table 7 confirms: random sampling sequences in biology is bound to overestimate results.

**AAV.** Table 5 summarizes baseline results for the biologically motivated AAV splits. Across all splits, purely supervised models are competitive with pretrained models. This suggests that the large sizes of training sets are past the threshold where pretraining improves performance. The particular choice of pooling that performs best is inconsistent across splits. The BLOSUM62 baseline could not be applied as the mutations in this set include insertions and deletions. In this case too, the sampled split reported separately in Table 7 strongly suggest that random sampling sequences in biology may lead to overestimated results.

**Thermostability.** Table 6 summarizes baseline results for thermostability. Pretrained models consistently outperform supervised models on this task, suggesting that this landscape is not yet past the threshhold where pretraining improves performance. Interestingly, the supervised baselines based on untrained ESM embeddings do better than either ridge or CNN. Mean over subset (mut mean) and BLOSUM62 are not applicable for the Meltome landscape as the sequences are not evolutionarily related.

Table 5: AAV baselines (metric: Spearman correlation)

| Model | Mut-Des | Des-Mut | 1-vs-rest | 2-vs-rest | 7-vs-rest | low-vs-high |
|---|---|---|---|---|---|---|
| ESM-1b (per AA) | 0.76 | — | 0.03 | 0.65 | 0.65 | 0.39 |
| ESM-1b (mean) | 0.63 | 0.59 | 0.04 | 0.26 | 0.46 | 0.18 |
| ESM-1b (mut mean) | 0.70 | 0.70 | 0.31 | 0.65 | 0.61 | 0.33 |
| ESM-1v (per AA) | 0.79 | — | 0.10 | 0.70 | 0.70 | 0.34 |
| ESM-1v (mean) | 0.55 | 0.44 | 0.18 | 0.16 | 0.45 | 0.20 |
| ESM-1v (mut mean) | 0.70 | 0.71 | 0.44 | 0.64 | 0.64 | 0.31 |
| ESM-untrained (per AA) | 0.56 | — | 0.18 | 0.22 | 0.42 | 0.08 |
| ESM-untrained (mean) | 0.27 | 0.34 | 0.01 | 0.14 | 0.22 | 0.22 |
| ESM-untrained (mut mean) | 0.62 | 0.64 | 0.26 | 0.16 | 0.56 | 0.24 |
| Ridge | 0.64 | 0.53 | 0.22 | 0.03 | 0.65 | 0.12 |
| CNN | 0.71 | 0.75 | 0.48 | 0.74 | 0.74 | 0.34 |
| Levenshtein | 0.60 | -0.07 | -0.11 | 0.57 | 0.53 | 0.25 |
| BLOSUM62 | NA | NA | NA | NA | NA | NA |

Table 6: Thermostability baselines (metric: Spearman correlation)

| Model | Mixed | Human | Human-Cell |
|---|---|---|---|
| ESM-1b (per AA) | 0.68 | 0.71 | 0.76 |
| ESM-1b (mean) | 0.68 | 0.70 | 0.75 |
| ESM-1b (mut mean) | NA | NA | NA |
| ESM-1v (per AA) | 0.65 | 0.77 | 0.78 |
| ESM-1v (mean) | 0.67 | 0.75 | 0.74 |
| ESM-1v (mut mean) | NA | NA | NA |
| ESM-untrained (per AA) | 0.44 | 0.44 | 0.46 |
| ESM-untrained (mean) | 0.36 | 0.48 | 0.49 |
| ESM-untrained (mut mean) | NA | NA | NA |
| Ridge | 0.17 | 0.15 | 0.24 |
| CNN | 0.34 | 0.50 | 0.49 |
| Levenshtein | NA | NA | NA |
| BLOSUM62 | NA | NA | NA |

## 6 Discussion

The prediction tasks in FLIP probe complex fitness landscapes across different protein functions. We curate three landscapes published in existing literature and formulate 15 corresponding splits of the data to mimic protein engineering tasks. The main criteria to include a landscape was whether it could be used to assess interesting types of generalization, and if it was amenable to interpretable assessment metrics. As no standard approach exists to partition landscapes arising from mutagenesis of a parent sequence, we propose ideas that may be applied to future landscapes. In particular, we explore the concept of training on sequences only a few mutations from a parent while predicting on data many mutations from a parent in a step-by-step fashion.

The need for more challenging splits is illustrated in Table 7, which shows results for the *sampled* splits, based on simple random sampling. Almost all models do drastically better for the sampled splits, and differences between models are exaggerated. This indicates the importance of biologically-motivated generalization in task design.

In general, results on baselines highlight that while pretraining approaches perform well on tasks with diverse sequences (Thermostability, Table 6), they do not outperform simpler models on mutational landscapes (GB1, Table 4 &, AAV, Table 5). In addition, large pretrained models require amounts of compute (up to 50 days on an NVidia A6000 GPU) to train on some tasks, which is out of the reach of most academic research groups. It is important to note that while we performed a modest hyperparameter search, more extensive sweeps combined with training data regularization

Table 7: Optimistic results for random splits (*Sampled)* on the AAV and GB1 sets (metric: Spearman correlation)

| Landscape | AAV | GB1 |
|---|---|---|
| ESM-1b (per AA) | 0.90 | 0.92 |
| ESM-1v (per AA) | 0.92 | 0.92 |
| ESM-untrained (per AA) | 0.78 | 0.79 |
| Ridge | 0.83 | 0.82 |
| CNN | 0.92 | 0.91 |

like different validation splits, may yield better absolute and relative performance. The landscapes and derived prediction splits offered in FLIP highlight directions for future work, such as better pretraining or embedding methods for protein mutational landscapes.

# 7    Conclusion

The proliferation of protein sequence data, along with advanced experimental techniques for functional measurement of proteins, presents a ripe environment for machine learning-enabled solutions in protein engineering. With the introduction of FLIP, we focus on sequence-fitness prediction and aim to encourage rigorous evaluation of model generalization in multiple tasks and settings relevant to protein engineering. We hope to seed advances in this emerging interdisciplinary field with downstream applications for solutions in human health and the environment. FLIP data and scripts are available under free licenses at https://benchmark.protein.properties.

## Acknowledgments and Disclosure of Funding

The authors thank Jeffrey Spencer, Sam Sinai, Sam Bowman, Roshan Rao and Debora Marks for ideas and discussions that helped us improve our work. The authors would also like to thank Helix and Murphy for careful attention to the manuscript. C.D. acknowledges support from the Bundesministerium für Bildung und Forschung (BMBF) – Project numbers: 01IS17049 and 031L0168. K.E.J. and B.J.W. acknowledge the NSF Division of Chemical, Bioengineering, Environmental and Transport Systems (1937902). N.B. was supported in part by NIH grant R35-GM134922 and by the Exascale Computing Project (17-SC-20-SC), a collaborative effort of the U.S. Department of Energy Office of Science and the National Nuclear Security Administration. S.G. thanks the MIT Machine Learning for Pharmaceutical Discovery and Synthesis Consortium for supporting this work. K.K.Y. was previously employed by Generate Biomedicines.

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
