## Glossary

**Epistasis** - in the most general sense, epistasis is interactions leading to non-independence of effects. For proteins, this means that the effect of a mutation in a protein sequence on fitness can vary based on co-occurring mutations.

**Fitness** - ability of a protein sequence to perform a specific, desired function.

**Fitness landscape** - both (1) a dataset mapping many protein sequences to fitness within a defined region of sequence space and (2) a conceptual framework for thinking about the mapping of protein sequence to fitness.

**Function** - a task performed by a protein sequence, typically referring to either a native task or a task desired by a protein engineer.

**Homology** - sharing a common origin at all levels (organism, population and species), which often results in similarity. For proteins both sequences and structures can be considered homologous due to common origin. [1]

**Multiple sequence alignment** - an arrangement of three or more sequences such that similar regions are aligned. Gaps can be inserted within some sequences at a penalty such that as much of the similar regions of the sequences are aligned as possible.

**Mutagenesis** - introduction of genetic mutations. In protein engineering, mutagenesis is typically performed on a single DNA sequence encoding a protein.

**Mutant** - a resulting DNA (and, equivalently, protein) sequence from mutagenesis on an initial starting sequence. Mutagenesis for protein engineering can either result in a single mutant or a library (pool) of mutants. Parent sequence - another word for the initial starting sequence prior to mutagenesis. This is not to be conflated with "wild type sequence".

**Sequence identity** - similarity between two (typically aligned) sequences

**Thermostability** - ability of a protein to preserve its structure and function under extremes of temperature conditions. [2]

**Tree of life** - referring to the phylogenetic tree of life, which depicts the relationships of biological species based on their last common ancestors.

**Variant** - within this text we define variant the same way as mutant (see previous).

**Wild type sequence** - a protein sequence that arises in nature and predominates within a natural population. While a wild type sequence can function as a parent sequence, these two terms have distinct meanings and should not be conflated.

## References

[1] Helga Ochoterena, Alexander Vrijdaghs, Erik Smets, and Regine Claßen-Bockhoff. The Search for Common Origin: Homology Revisited. *Systematic Biology*, 68(5):767–780, 02 2019. ISSN 1063-5157. doi: 10.1093/sysbio/syz013. URL https://doi.org/10.1093/sysbio/syz013. (document)

[2] X-X Zhou, Y-B Wang, Y-J Pan, and W-F Li. Differences in amino acids composition and coupling patterns between mesophilic and thermophilic proteins. *Amino acids*, 34(1):25–33, 2008. (document)

# GB1 Additional Results [*]

| Model | Split | Train rho | Train MSE | Test rho | Test MSE | Avg Epochs Trained |
|---|---|---|---|---|---|---|
| ESM-1b (per AA) | low_vs_high | 0.83 | 0.03 | 0.53 | 3.26 | 60.00 |
| | one_vs_rest | 0.58 | 0.79 | 0.29 | 1.83 | 22.73 |
| | sampled | 0.92 | 0.24 | 0.91 | 0.28 | 69.27 |
| | three_vs_rest | 0.76 | 0.58 | 0.79 | 0.86 | 36.45 |
| | two_vs_rest | 0.59 | 0.60 | 0.47 | 1.41 | 56.64 |
| ESM-1b (mean) | low_vs_high | 0.75 | 0.09 | 0.13 | 4.33 | 41.00 |
| | one_vs_rest | 0.56 | 0.84 | 0.31 | 1.88 | 22.00 |
| | sampled | 0.71 | 1.20 | 0.68 | 1.18 | 31.27 |
| | three_vs_rest | 0.36 | 1.12 | 0.54 | 1.59 | 22.64 |
| | two_vs_rest | 0.38 | 0.99 | 0.35 | 1.58 | 24.27 |
| ESM-1b (mut mean) | low_vs_high | 0.83 | 0.03 | 0.48 | 3.39 | 48.45 |
| | one_vs_rest | 0.68 | 0.54 | 0.27 | 1.67 | 26.00 |
| | sampled | 0.92 | 0.31 | 0.89 | 0.40 | 61.27 |
| | three_vs_rest | 0.81 | 0.50 | 0.80 | 0.88 | 37.36 |
| | two_vs_rest | 0.69 | 0.50 | 0.55 | 1.28 | 65.00 |
| ESM-1v (per AA) | low_vs_high | 0.82 | 0.03 | 0.53 | 3.26 | 54.73 |
| | one_vs_rest | 0.65 | 0.70 | 0.09 | 1.92 | 25.00 |
| | sampled | 0.94 | 0.17 | 0.92 | 0.22 | 70.91 |
| | three_vs_rest | 0.87 | 0.32 | 0.83 | 0.71 | 54.64 |
| | two_vs_rest | 0.65 | 0.48 | 0.37 | 1.48 | 55.36 |
| ESM-1v (mean) | low_vs_high | 0.74 | 0.09 | 0.11 | 4.54 | 42.55 |
| | one_vs_rest | 0.65 | 0.79 | -0.08 | 1.88 | 24.73 |
| | sampled | 0.73 | 1.12 | 0.69 | 1.13 | 34.27 |
| | three_vs_rest | 0.42 | 1.07 | 0.49 | 1.53 | 23.27 |
| | two_vs_rest | 0.40 | 0.92 | 0.18 | 1.54 | 29.36 |
| ESM-1v (mut mean) | low_vs_high | 0.82 | 0.03 | 0.47 | 3.50 | 54.36 |
| | one_vs_rest | 0.68 | 0.48 | 0.19 | 2.13 | 24.55 |
| | sampled | 0.93 | 0.27 | 0.90 | 0.35 | 66.36 |
| | three_vs_rest | 0.80 | 0.53 | 0.80 | 0.95 | 35.36 |
| | two_vs_rest | 0.68 | 0.51 | 0.50 | 1.34 | 54.18 |
| ESM-untr (per AA) | low_vs_high | 0.79 | 0.04 | 0.32 | 3.78 | 77.91 |
| | one_vs_rest | -0.08 | 0.84 | 0.12 | 1.88 | 24.18 |
| | sampled | 0.85 | 0.54 | 0.81 | 0.57 | 119.27 |
| | three_vs_rest | 0.43 | 1.11 | 0.52 | 1.55 | 23.36 |
| | two_vs_rest | 0.00 | 1.00 | 0.22 | 1.58 | 29.18 |
| ESM-untr (mean) | low_vs_high | 0.61 | 0.10 | 0.10 | 4.71 | 54.18 |
| | one_vs_rest | -0.17 | 0.84 | 0.05 | 1.82 | 25.64 |
| | sampled | 0.55 | 1.45 | 0.55 | 1.40 | 30.27 |
| | three_vs_rest | 0.33 | 1.19 | 0.46 | 1.69 | 22.64 |
| | two_vs_rest | -0.10 | 1.00 | 0.09 | 1.57 | 30.00 |
| ESM-untr (mut mean) | low_vs_high | 0.73 | 0.06 | 0.16 | 4.14 | 63.55 |
| | one_vs_rest | 0.44 | 0.95 | 0.21 | 1.62 | 22.00 |
| | sampled | 0.73 | 0.82 | 0.70 | 0.85 | 72.27 |
| | three_vs_rest | 0.48 | 0.93 | 0.57 | 1.26 | 25.00 |
| | two_vs_rest | 0.51 | 0.70 | 0.40 | 1.63 | 142.18 |
| CNN | low_vs_high | 0.84 | 0.03 | 0.47 | 3.40 | 99.00 |
| | one_vs_rest | 0.48 | 2.52 | 0.15 | 2.11 | 22.73 |
| | sampled | 0.91 | 0.24 | 0.90 | 0.30 | 99.00 |
| | three_vs_rest | 0.83 | 0.40 | 0.81 | 0.75 | 99.00 |
| | two_vs_rest | 0.58 | 1.05 | 0.39 | 1.64 | 28.09 |

* All additional result tables are averages of 10 separate runs with different seeds; graphs show mean and standard dev

# AAV Additional Results

| Model | Split | Train rho | Train MSE | Test rho | Test MSE | Avg Epochs Trained |
|---|---|---|---|---|---|---|
| ESM-1b (per AA) | low_vs_high | 0.78 | 1.15 | 0.38 | 14.52 | 49.67 |
| | one_vs_many | 0.38 | 5.73 | 0.03 | 9.85 | 29.73 |
| | sampled | 0.90 | 2.07 | 0.90 | 2.12 | 44.27 |
| | seven_vs_many | 0.89 | 2.56 | 0.65 | 2.70 | 43.82 |
| | two_vs_many | 0.84 | 2.42 | 0.61 | 6.79 | 67.45 |
| ESM-1b (mean) | low_vs_high | 0.59 | 2.03 | 0.15 | 19.69 | 38.73 |
| | one_vs_many | 0.38 | 5.70 | 0.04 | 9.78 | 31.09 |
| | sampled | 0.78 | 5.16 | 0.78 | 5.19 | 35.09 |
| | seven_vs_many | 0.76 | 6.24 | 0.46 | 5.96 | 26.91 |
| | two_vs_many | 0.73 | 5.07 | 0.22 | 9.74 | 42.82 |
| ESM-1b (mut mean) | low_vs_high | 0.74 | 1.49 | 0.32 | 15.54 | 32.64 |
| | one_vs_many | 0.66 | 3.96 | 0.40 | 13.86 | 49.09 |
| | sampled | 0.89 | 2.62 | 0.88 | 2.70 | 33.45 |
| | seven_vs_many | 0.88 | 3.40 | 0.60 | 4.51 | 27.64 |
| | two_vs_many | 0.82 | 3.12 | 0.59 | 7.41 | 34.73 |
| ESM-1v (per AA) | low_vs_high | 0.78 | 1.11 | 0.35 | 10.83 | 39.91 |
| | one_vs_many | 0.30 | 5.77 | 0.12 | 9.55 | 22.82 |
| | sampled | 0.91 | 1.88 | 0.91 | 1.94 | 37.64 |
| | seven_vs_many | 0.91 | 2.22 | 0.70 | 2.58 | 33.55 |
| | two_vs_many | 0.87 | 1.82 | 0.70 | 6.00 | 50.82 |
| ESM-1v (mean) | low_vs_high | 0.58 | 2.05 | 0.23 | 22.12 | 38.45 |
| | one_vs_many | 0.34 | 5.77 | 0.15 | 9.56 | 23.36 |
| | sampled | 0.78 | 5.30 | 0.77 | 5.29 | 30.91 |
| | seven_vs_many | 0.76 | 6.03 | 0.45 | 8.85 | 26.73 |
| | two_vs_many | 0.71 | 4.98 | 0.14 | 12.14 | 37.27 |
| ESM-1v (mut mean) | low_vs_high | 0.75 | 1.50 | 0.29 | 16.14 | 29.27 |
| | one_vs_many | 0.67 | 4.40 | 0.43 | 18.06 | 39.82 |
| | sampled | 0.89 | 2.94 | 0.89 | 3.03 | 30.27 |
| | seven_vs_many | 0.89 | 3.42 | 0.63 | 4.35 | 25.00 |
| | two_vs_many | 0.84 | 2.54 | 0.61 | 8.19 | 40.64 |
| | low_vs_high | 0.56 | 1.78 | 0.06 | 23.25 | 62.73 |
| | one_vs_many | 0.40 | 5.79 | 0.18 | 9.54 | 25.91 |
| | sampled | 0.78 | 4.15 | 0.77 | 4.25 | 100.36 |
| | seven_vs_many | 0.74 | 4.70 | 0.38 | 15.27 | 56.18 |
| | two_vs_many | 0.73 | 4.32 | 0.20 | 11.02 | 118.27 |
| ESM-untr (mean) | low_vs_high | 0.42 | 2.22 | 0.22 | 25.52 | 70.18 |
| | one_vs_many | 0.32 | 5.80 | 0.01 | 9.54 | 25.73 |
| ESM-untr (per AA) | sampled | 0.62 | 6.55 | 0.62 | 6.58 | 69.73 |
| | seven_vs_many | 0.62 | 6.33 | 0.23 | 18.30 | 44.36 |
| | two_vs_many | 0.57 | 6.15 | 0.14 | 9.15 | 90.27 |
| ESM-untr (mut mean) | low_vs_high | 0.63 | 1.64 | 0.24 | 19.26 | 47.91 |
| | one_vs_many | 0.52 | 4.54 | 0.41 | 8.66 | 112.73 |
| | sampled | 0.81 | 3.70 | 0.81 | 3.79 | 68.55 |
| | seven_vs_many | 0.80 | 3.80 | 0.56 | 7.40 | 60.27 |
| | two_vs_many | 0.75 | 4.25 | 0.18 | 8.92 | 49.36 |
| CNN | low_vs_high | 0.80 | 1.03 | 0.28 | 15.41 | 99.00 |
| | one_vs_many | 0.65 | 5.78 | 0.35 | 9.53 | 37.18 |
| | sampled | 0.92 | 1.51 | 0.91 | 1.70 | 99.00 |
| | seven_vs_many | 0.92 | 1.44 | 0.73 | 5.00 | 99.00 |
| | two_vs_many | 0.86 | 1.74 | 0.58 | 10.39 | 99.00 |

**Thermostability Additional Results**

| Model | Split | Train rho | Train MSE | Test rho | Test MSE | Avg Epochs Trained |
|---|---|---|---|---|---|---|
| ESM-1b (per AA) | human | 0.76 | 9.54 | 0.71 | 11.98 | 33.9 |
| | human_cell | 0.76 | 13.60 | 0.68 | 18.01 | 38.2 |
| | mixed_split | 0.74 | 25.72 | 0.67 | 39.86 | 35.2 |
| ESM-1b (mean) | human | 0.75 | 9.97 | 0.70 | 11.98 | 36.0 |
| | human_cell | 0.74 | 14.43 | 0.67 | 17.94 | 40.8 |
| | mixed_split | 0.72 | 27.38 | 0.67 | 39.87 | 44.0 |
| ESM-1v (per AA) | human | 0.77 | 8.96 | 0.69 | 12.59 | 32.8 |
| | human_cell | 0.78 | 12.37 | 0.67 | 18.16 | 36.5 |
| | mixed_split | 0.76 | 22.16 | 0.65 | 41.12 | 33.2 |
| ESM-1v (mean) | human | 0.75 | 9.61 | 0.69 | 12.59 | 36.6 |
| | human_cell | 0.76 | 13.44 | 0.67 | 18.09 | 43.0 |
| | mixed_split | 0.73 | 25.55 | 0.66 | 40.28 | 37.2 |
| ESM-untr (per AA) | human | 0.45 | 18.33 | 0.48 | 18.85 | 51.2 |
| | human_cell | 0.47 | 25.01 | 0.48 | 25.23 | 59.4 |
| | mixed_split | 0.41 | 54.94 | 0.45 | 61.79 | 131.5 |
| ESM-untr (mean) | human | 0.48 | 17.64 | 0.52 | 17.84 | 52.2 |
| | human_cell | 0.49 | 24.31 | 0.52 | 24.30 | 60.6 |
| | mixed_split | 0.34 | 62.60 | 0.37 | 71.80 | 137.0 |
| CNN | human | 0.68 | 12.41 | 0.51 | 18.02 | 77.5 |
| | human_cell | 0.71 | 16.73 | 0.49 | 25.36 | 83.4 |
| | mixed_split | 0.73 | 27.89 | 0.34 | 84.49 | 99.0 |

# GB1 Dataset

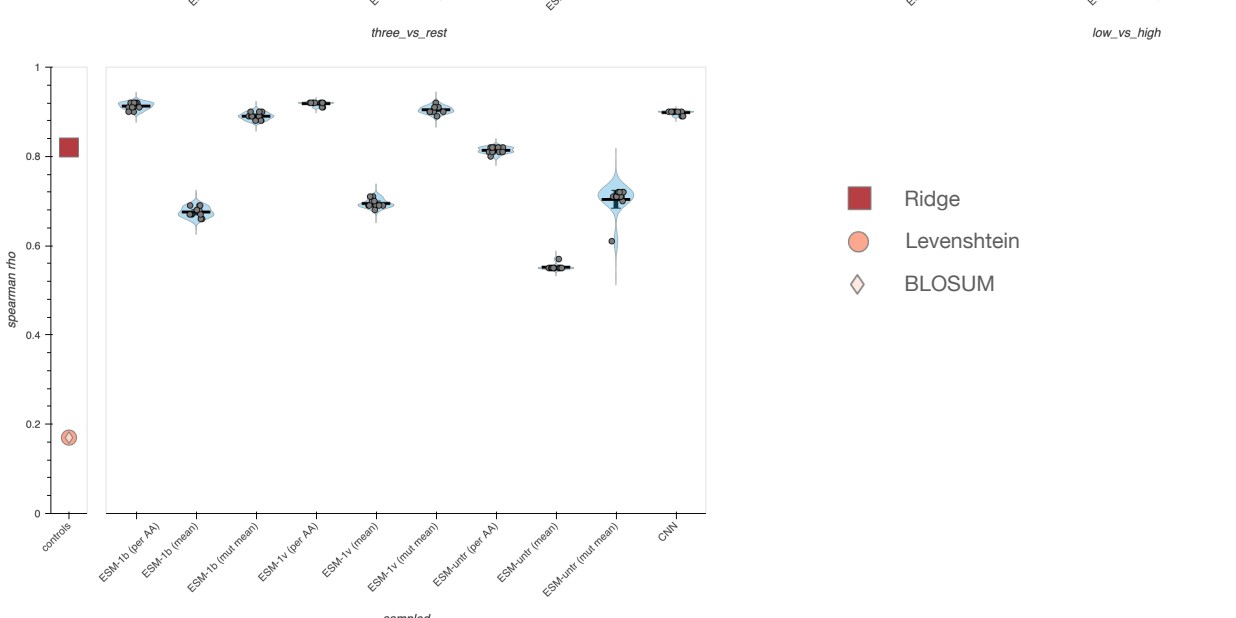

Legend:
- Ridge
- Levenshtein
- BLOSUM

# AAV Dataset

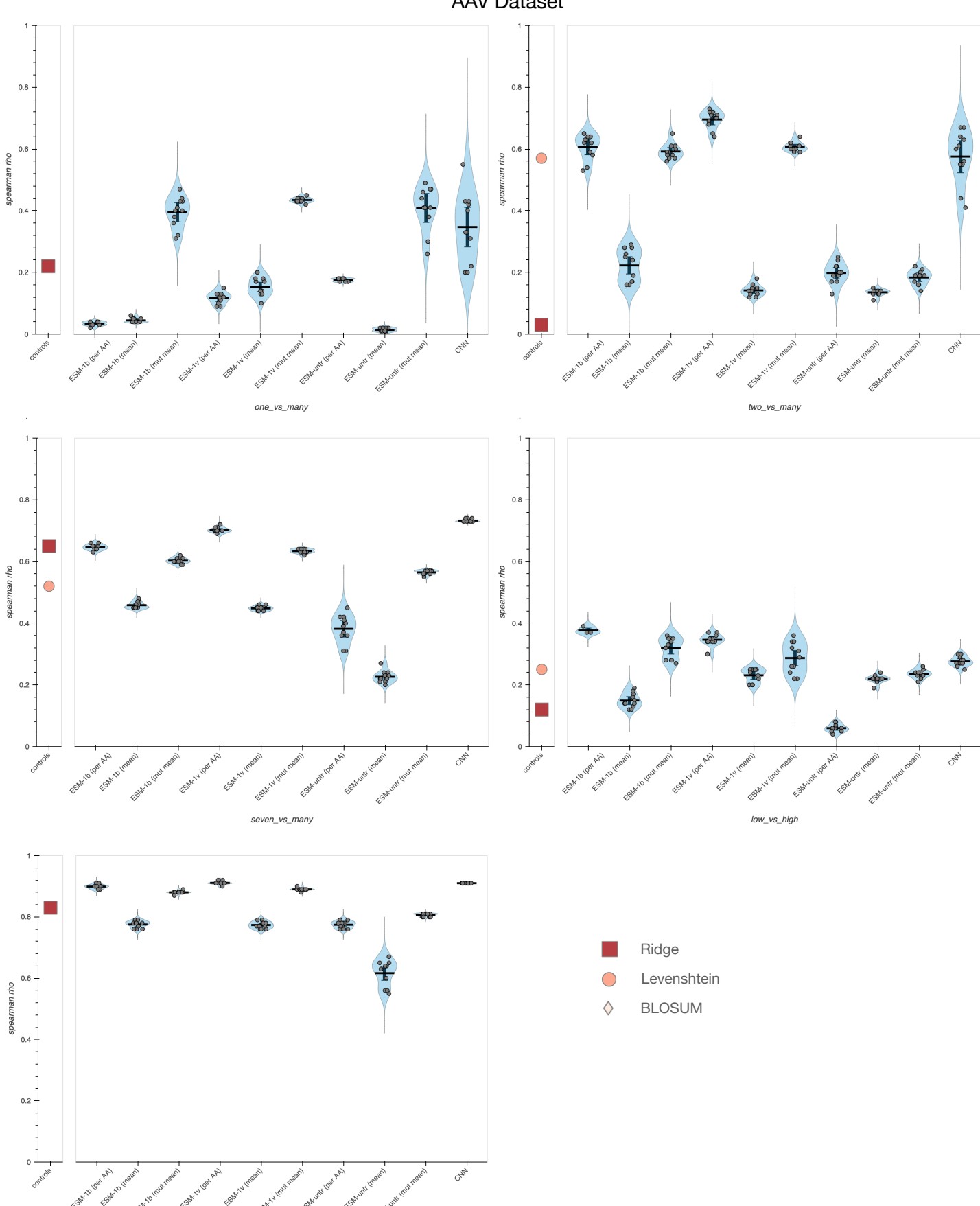

# Thermostability Dataset

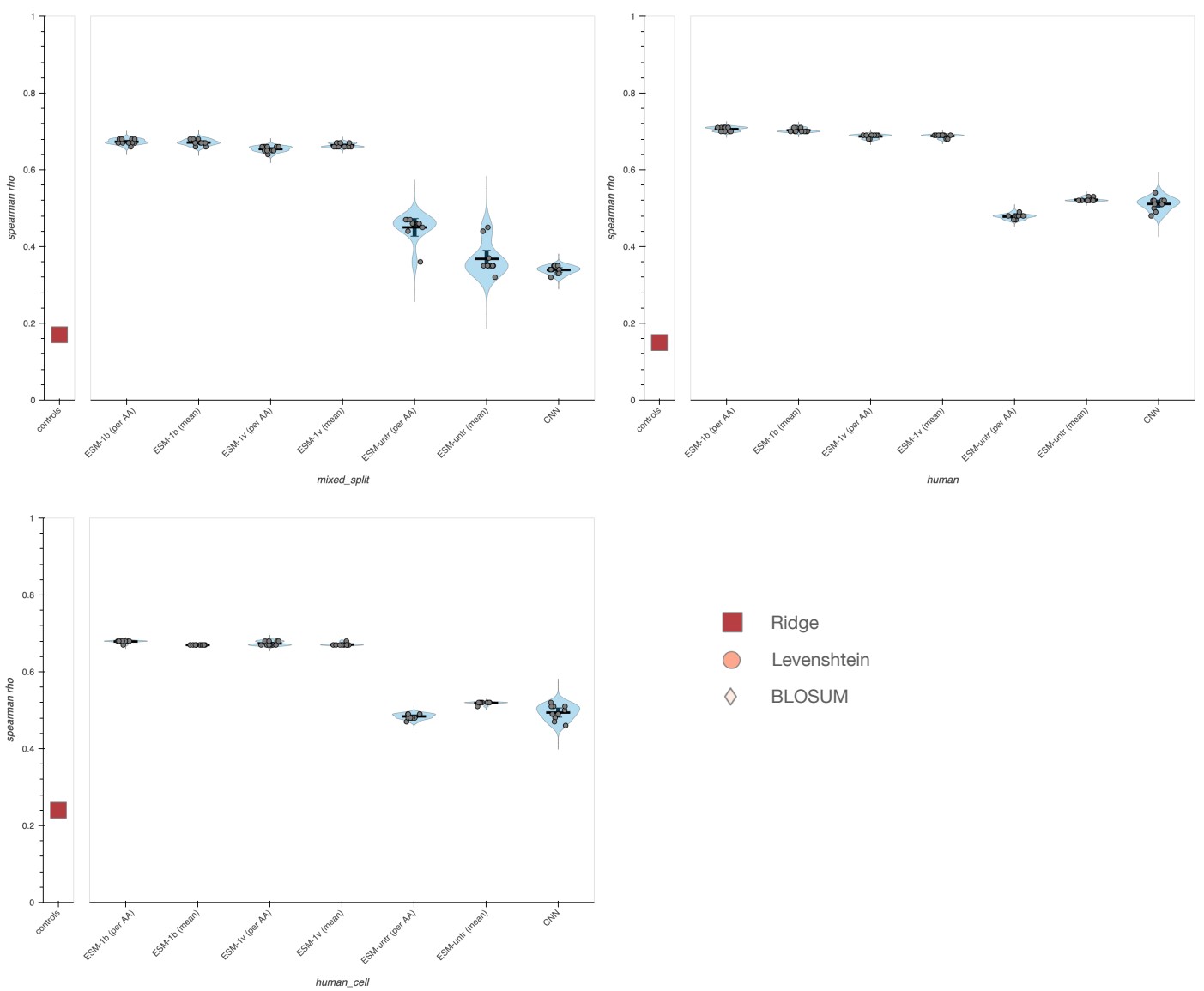

Ridge

Levenshtein

BLOSUM