# OpenReview forum: "FLIP: Benchmark tasks in fitness landscape inference for proteins"
_NeurIPS.cc/2021/Track/Datasets_and_Benchmarks/Round2 — NeurIPS 2021 Datasets and Benchmarks Track (Round 2)_

### Official Review · Reviewer_e4HJ · 2021-09-12
**FLIP Standardizes Protein Fitness Prediction for the Machine Learning Community**

**Rating:** 7
**Confidence:** 4
**Clarity:** This reviewer believes the manuscript…

**Strengths:**

FLIP takes a great leap forward towards standardizing protein fitness prediction for the machine learning community via its biologically-informed tasks and cross-validation dataset splits. By incorporating important biological insights into the design of their datasets and benchmarks, the authors have developed research infrastructure for accelerated representation learning of protein sequences. Having more clearly defined benchmarks for protein fitness could enable more directed efforts into engineering new proteins to, for example, help address climate change or target existing diseases.

**Weaknesses:**

The main weakness this reviewer can identify is in some of the decision statements made by the authors.

For example, in Section 3 (Landscape and Tasks), the authors say "We prioritized complex landscapes rather than single amino acid variants, allowing for a larger potential sequence space as well as functional diversity to ensure model generalization for broad applicability". Are there any references to support this decision/conclusion?

Another example comes from Section 6 (Discussion): "It is important to note that we do not perform hyperparameter search for any baseline method. However, given that applied users often accept defaults, this is a meaningful comparison." To this reviewer, this claim seems a bit overly general. Do the authors have any specific instances/examples to give in which certain applied users (e.g., biologists) may be comfortable accepting default model arguments (as opposed to instances in which they may not)?

**Additional Feedback:**

This reviewer would like to thank the authors for their diligent efforts in standardizing protein fitness prediction for the machine learning community. This reviewer believes such work may have great broader impacts resulting from it, such as accelerated biological and representation learning research for climate change.

**Correctness:**

This reviewer cannot identify any improperly designed benchmarks or incorrect claims made by the authors. The datasets themselves also appear to be properly formatted for users' benefit.

In terms of small typing errors in the manuscript, an unnecessary space can be found in Section 7 (Conclusion) in the phrase "With the introduction of FLIP , we...".

**Documentation:**

This reviewer believes the documentation for the authors' datasets and benchmarks is clear.

**Ethics:**

The work described by the authors entails fundamental biophysical research driven by machine learning. As such, I do not see any direct societal concerns resulting from this research.

**Relation To Prior Work:**

This reviewer believes the authors clearly outline the position of their work in relation to previous contributions.

**Summary And Contributions:**

FLIP curates transparent datasets, tasks, and benchmarks for protein fitness prediction. In addition, the authors include a keen analysis of which representation learning algorithms may be the best way forward towards increased performance for each fitness prediction task.

---

> ### Author Response · Authors · 2021-09-30
> **Response**
>
> > For example, in Section 3 (Landscape and Tasks), the authors say "We prioritized complex landscapes rather than single amino acid variants, allowing for a larger potential sequence space as well as functional diversity to ensure model generalization for broad applicability". Are there any references to support this decision/conclusion?
>
> Good point. There is some evidence that deep mutational scans contain noise (Marquet et al, 2021), (full disclosure, C.D. is a co-author on that and initiated the work), and thus these were deprioritized for round 0. However, to our knowledge, papers analyzing this in detail are not published (as a matter of fact: one of the team is working on a manuscript describing just that). In either case, in the absence of clear scientific backing, we toned it down!
>
> > Another example comes from Section 6 (Discussion): "It is important to note that we do not perform hyperparameter search for any baseline method. However, given that applied users often accept defaults, this is a meaningful comparison." To this reviewer, this claim seems a bit overly general. Do the authors have any specific instances/examples to give in which certain applied users (e.g., biologists) may be comfortable accepting default model arguments (as opposed to instances in which they may not)?
>
> We did change this sentence and there’s some discussion around this topic in other responses above. I (CD) do however feel that you are bringing up an interesting “general” point, which I’m happy to give you my 2c on. I (co-)developed several bioinformatics tools (latest: bioembeddings.com), many of which were turned to online servers (evcouplings.org, predictprotein.org, embed.protein.properties), conservatively used by around 3k users/mo. If there is one thing I can guarantee is that the majority of users will not bother with selecting (or even understanding) parameters: they will accept that they (defaults) are “good enough” or even “good”! For ML, I’ve seen: if they are given a certain training recipe, they will apply it as-is (https://doi.org/10.1002/cpz1.113). This is why in evcouplings we offer to produce a few alignments with different bitscore cutoffs by tweaking the most sensitive parameter in the hope that at least one alignment will be good. Predictprotein used to provide many tweakable parameters, but after overwhelming confusion, it’s now a simple: paste in sequence + press submit.
>
> I don’t know of any paper that studies this particular issue (“do people take ML pipelines and apply them 1-to-1 to their problem?”), but you are absolutely right: we *should* dig deeper here (note: I would claim this is more behavioural science than ML), and if it doesn’t exist, it may very well be an interesting manuscript. The dilemma to me is: make tools easy/obvious/... so that everyone feels good using them and can get started (downside: maybe use them the wrong way!) vs. put checks and balances in (downside: frustrating).
>
> > In terms of small typing errors in the manuscript, an unnecessary space can be found in Section 7 (Conclusion) in the phrase "With the introduction of FLIP , we...".
>
> Thank you for spotting. We fixed this issue.

---

### Official Review · Reviewer_DPa5 · 2021-09-20
**Interesting Benchmark Addressing a Niche in Protein Function Prediction**

**Rating:** 7
**Confidence:** 3

**Strengths:**

&nbsp;

1. The benchmark is **carefully designed** leveraging domain knowledge of protein engineering e.g. biologically relevant splits and choice of fitness landscapes for inclusion in the benchmark. The attention given to low-N tasks i.e. low-resource tasks is particularly appealing, a situation which arises in many areas of the experimental sciences [1] with few exceptions [2]!

2. The benchmark is **complementary to well-known benchmarks** in the ML community such as CASP which focus on protein structure prediction. In particular, the paper also serves as an accessible guide to ML researchers interested in protein engineering applications.

&nbsp;


**Weaknesses:**

&nbsp;

## __MAJOR POINTS__

&nbsp;

1. The model hyper-parameters for the benchmark are not tuned. I appreciate that some models require a lot of compute and the purpose of this study is to demonstrate how the benchmark might be used, however I would suggest **modifying some of the claims made in the paper where applicable**. In particular, the absence of hyper-parameter tuning should preclude the use of appropriate statistical hypothesis tests for validating claims. I highlight some of the statements:

2. *pretraining offered by ESM-1b [34] or ESM-1v [16] does not help much when sufficient supervised data is available.* Is this statement a defensible conclusion if the models have not been tuned?

3. Regarding the statement in the caption of Table 5, *This suggests that the large sizes of training sets are past the threshold where pretraining improves performance*. Is this a valid conclusion if the models (including ridge regression presumably) have not been tuned?

4. Regarding the statement in the conclusion, *However, given that applied users often accept defaults, this is a meaningful comparison.* I contend that this is a valid statement. I think it should be removed and the conclusions from the baseline comparison should be amended.

5. Perhaps some **guidance on how practitioners would use the benchmark** in the scope of a real-world pipeline would be desirable i.e. how do ML models actually benefit laboratory scientists in the context of designing a protein with a desired function? The authors should have an extra two pages for the camera-ready version (the submitted version page limit is actually 9 pages, instead of 8, and there is an extra page made available for the camera-ready version!).

6. A **glossary of terms** defining concepts such as epistatic, tree of life, single amino acid mutations, wild-type to a lay ML audience may be desirable.

7. It would be nice to **introduce the task names before they are used in Table 2.**

8. It would be beneficial if the **codebase documentation were improved** e.g. the models.py module contains the home directory of one of the authors. It would be nice if the README.md contained an example of evaluating a custom model on the benchmark task.

&nbsp;

## __MINOR POINTS__

&nbsp;

1. "account" instread of "accounting" in the Related Work section on page 3.

2. The results tables are challenging to interpret because of a lack of boldface in the results figures e.g. in Table 1 - what exactly is the conclusion for the table?

3. Table 4 and 5 captions, it would be nice if the reported metric was given here.

4. It may be an irrelevant point but is there any benefit to representing proteins as graphs in the benchmark? [3].

&nbsp;

## __REFERENCES__

&nbsp;

1. Thawani et al. The photoswitch dataset: A molecular machine learning benchmark for the advancement of synthetic chemistry. arXiv. 2020.

2. Hey et al. Machine learning and big scientific data. Philosophical Transactions of the Royal Society A. 2020.

3. Jamasb et al. Graphein - A Python library for geometric deep learning and network analysis on protein structures. bioRxiv. 2020.

4. Gebru et al. Datasheets for datasets. arXiv. 2018.

&nbsp;

**Additional Feedback:**

&nbsp;

All points covered in the main response.

&nbsp;

**Clarity:**

&nbsp;

The paper is well written and in general accessible to a lay machine learning audience. Improvements suggested above.

&nbsp;

**Correctness:**

&nbsp;

To the best of my knowledge the claims made by the authors are correct save for those identified in the comments above.

&nbsp;

**Documentation:**

&nbsp;

There is scope for improving the documentation of the codebase as well as the datasets. In particular the **inclusion of a datasheet** [4] would help. Apologies if I have missed this in the codebase, the README might be a good location to direct users to the datasheet specification.

&nbsp;

**Ethics:**

&nbsp;

Adequately addressed.

&nbsp;

**Relation To Prior Work:**

&nbsp;

Adequately addressed.

&nbsp;

**Summary And Contributions:**

&nbsp;

The paper presents Fitness Landscape Inference for Proteins (FLIP), a benchmark for protein function prediction. Catering for relevant real-world protein engineering settings such as low-resource makes the benchmark especially appealing given the scarcity of high-quality experimental data in general in the natural sciences.

I believe the dataset is an interesting and novel contribution to the machine learning community and as such I lean on the side of acceptance. In particular, I think it is important to introduce machine learning practitioners to novel applications such as protein engineering. There is a synergistic interaction between new data formats and ML methodology and such benchmarks are liable to pave the way for advances in core ML, a theme very much in keeping with the directive of the Datasets and Benchmarks track.

I have a few concerns which, if addressed in the rebuttal, may warrant an increased score.

&nbsp;

---

> ### Author Response · Authors · 2021-09-30
> **Response pt 1**
>
> > The model hyper-parameters for the benchmark are not tuned. I appreciate that some models require a lot of compute and the purpose of this study is to demonstrate how the benchmark might be used, however I would suggest modifying some of the claims made in the paper where applicable. In particular, the absence of hyper-parameter tuning should preclude the use of appropriate statistical hypothesis tests for validating claims. I highlight some of the statements:
>
> A very consistent criticism taken! We refer you to previous responses on the matter.
>
> > pretraining offered by ESM-1b [34] or ESM-1v [16] does not help much when sufficient supervised data is available. Is this statement a defensible conclusion if the models have not been tuned?
>
> Considering “tuning” of the models on top of the embeddings rather than the embedders themselves the answer remains: yes. We have clarified in text that this (frozen embeddings, tuning on top) is the correct interpretation of that statement.
>
> > Regarding the statement in the caption of Table 5, This suggests that the large sizes of training sets are past the threshold where pretraining improves performance. Is this a valid conclusion if the models (including ridge regression presumably) have not been tuned?
>
> Similar to above + we have tested three learning rates on our 1D attention and linear layers; while the overall results vary, the trend between the supervised ESM models, randomly initialized ESM models, and non-pretrianed models stay the same.
>
> > Regarding the statement in the conclusion, However, given that applied users often accept defaults, this is a meaningful comparison. I contend that this is a valid statement. I think it should be removed and the conclusions from the baseline comparison should be amended.
>
> Point taken and amended! Specifics in the new version of manuscript and an answer to reviewer two.
>
> > Perhaps some guidance on how practitioners would use the benchmark in the scope of a real-world pipeline would be desirable i.e. how do ML models actually benefit laboratory scientists in the context of designing a protein with a desired function? The authors should have an extra two pages for the camera-ready version (the submitted version page limit is actually 9 pages, instead of 8, and there is an extra page made available for the camera-ready version!).
>
>
> This is an excellent point. In fact, we designed some of our splits to reflect standard laboratory data collection procedures. In other words, some of our splits are designed to test model generalizability, while others are designed to test model applicability (when considering existing data-collection strategies/technologies). We have made this fact clearer in the Landscapes and Splits section as well as while introducing each of the splits considered. The 1-vs-rest and 2-vs-rest splits are noted as being the most generally practically applicable.
>
> > A glossary of terms defining concepts such as epistatic, tree of life, single amino acid mutations, wild-type to a lay ML audience may be desirable.
>
> This was a very nice suggestion! We have added a glossary of terms to our supplement as well as mention of the glossary in the text. Our glossary highlights some of the protein engineering terms that an ML audience may not know, including those recommended.
>
> > It would be nice to introduce the task names before they are used in Table 2.
>
> We played around with this a bit - putting the table after the three sections where we introduce and describe splits, or before (as it was/remains). We feel that this way, those who want to quickly get a sense of what is state of the art can do so without immediately engaging in details. This increases our ability to attract a wider range of ML practitioners to engage with FLIP. In fact, the “introduction” to Landscapes and Splits serves the double purpose of introduction and summary, the remaining bit is for you: the eager readers. We remain open to more feedback on this point!
>
> > It would be beneficial if the codebase documentation were improved e.g. the models.py module contains the home directory of one of the authors. It would be nice if the README.md contained an example of evaluating a custom model on the benchmark task.
>
> Whoops, the home dir escaped! We fixed that, thank you! Equally, we included some instructions in the README on how to reproduce the evaluation we reported in the manuscript, which should seed interested practitioners in getting started on their own evaluation.
>
> > "account" instread of "accounting" in the Related Work section on page 3.
>
> Fixed. Thank you!

---

> > ### Author Response · Authors · 2021-09-30
> > **Response pt 2/2**
> >
> > > The results tables are challenging to interpret because of a lack of boldface in the results figures e.g. in Table 1 - what exactly is the conclusion for the table?
> >
> > Ok: good points, we thought about this, too! Should we crown a winner? We think the answer is “no”, as statistical error marks the winner only in spirit, not in practice. In fact, we originally had boldface and removed it before submission. So, what’s the point? We do think it’s valuable to assess a few baselines + give our numbers in case anyone tries to reproduce + it helps us make some arguments in the text (e.g. does pre-training always help/is it always de-facto better? How are sampled splits vs. challenging splits?). True: lots of numbers, difficult to parse, where should one look,... but moving them to the appendix may be equally as frustrating for those that want to pair numbers to claims. Difficult one! For now: we left it as is.
> >
> > > Table 4 and 5 captions, it would be nice if the reported metric was given here.
> >
> > Thanks! Added.
> >
> > > It may be an irrelevant point but is there any benefit to representing proteins as graphs in the benchmark? [3].
> >
> > Not irrelevant, very interesting gedankenspiel, especially in the post AlphaFold2 running on Colab era. The challenge with 3D structure is that for most proteins this type of information is unavailable, and thus, confining predictors to predict for sequences where 3D is available would reduce the space of possibilities vastly. However, it may be the right time to attempt exercises deemed “too much” even just a year ago. Nevertheless, a few considerations remain even if we assume good-enough predicted structures: while there is a clear link between structure and function, small changes in structure can lead to large changes in function (a protein with n binding sites may lose one, or a protein with a disordered region that serves to energy minimize a binding process loses that ability). Interesting thread on this: https://twitter.com/LindorffLarsen/status/1440568064542609414, which came as a response to some researchers attempting to predict structures for mutants, with some very interesting discussions that followed: https://twitter.com/ewanbirney/status/1440244587499384839. Ultimately: interesting question, but beyond the scope of this benchmark.

---

> > > ### Comment · Reviewer_DPa5 · 2021-09-30
> > > **Many Thanks for the Response!**
> > >
> > > &nbsp;
> > >
> > > Many thanks for the authors for clarifying the nature of the train/test splitting procedure and its applications to real-world problems. The included glossary of terms should help to make the subject matter more accessible to a general machine learning audience! The results of the hyperparameter sweep are interesting, would this imply that the model defaults have been tuned in previous work? (If indeed they perform best on the new tasks considered here?). The amendments to the claims in the text are also very much appreciated!
> > >
> > > As mentioned in the response to the summary comment, a more detailed response will follow together with a re-evaluation of the paper in light of the authors' modifications!
> > >
> > > &nbsp;

---

> > > > ### Author Response · Authors · 2021-09-30
> > > > **Quick reply**
> > > >
> > > > THANKS. Due to T-1 min; short answer:
> > > >
> > > > > The results of the hyperparameter sweep are interesting, would this imply that the model defaults have been tuned in previous work? (If indeed they perform best on the new tasks considered here?).
> > > >
> > > >  ESM-1v was tuned on predicting variation; ESM-1b was not.

---

> > > > > ### Comment · Reviewer_DPa5 · 2021-10-04
> > > > > **Thanks for the Quick Clarification**
> > > > >
> > > > > &nbsp;
> > > > >
> > > > > Many thanks for the quick clarification, it was much appreciated!
> > > > >
> > > > > &nbsp;

---

### Official Review · Reviewer_WRMr · 2021-09-20
**A benchmark addressing a meaningful problem in ML-assisted protein biology**

**Rating:** 5
**Confidence:** 4
**Correctness:** The work seems correct other than the…

**Strengths:**

* The need for a meaningful benchmark in this context is timely, well-motivated and has the potential to be highly impactful as it is complementary to emerging benchmarks in computational protein science.

* The work presents a collection of tasks assessing fitness in a range of biological contexts/experimental setups covering the main relevant problem scenarios.

* The authors provide well-motivated splitting strategies for the datasets.

* The authors make the data curation open and accessible.

Overall I am positive about the direction of the work and its central premise. I am open to changing my score based on dialogue with the authors.

**Weaknesses:**

## Major
The majority of these comments stem from the view that benchmarks should be authoritative and complete if they are to serve as the basis of assessment by the community. From this perspective, I believe there is a little more work to be done to achieve this for FLIP. I reiterate that I am positive about FLIP and would like to see it adopted by the community but I think it falls short of this standard in its current form.

Hyperparameter tuning is not performed for baselines. The authors state they expect absolute and relative performance are likely to change following a hyperparameter search which makes it hard justify accepting the empirical results presented. The authors justify this as being a meaningful comparison because "applied users often accept defaults". This seems like a weak justification as a different set of hyperparameters could alter the findings and thus the direction of follow-on research if users were to simply accept defaults.  The purpose of benchmarks is to understand the relative merits of different modelling approaches on a relevant problem space and at present the authors do not present this. Furthermore, I think as a benchmark the authors should provide principled validation splits as this would be highly desirable by users of the benchmark.

I'm not sure I'm convinced that the inclusion of the meltome dataset is especially congruent with the other tasks in its current form. The other tasks focus on mutations along a single starting sequence and effectively sample large portions of the sequence space within a small number of edits. This allows for rich and meaningful characterisation of the fitness landscape. I appreciate the goal of including the meltome task is to increase the diversity of tasks to include a landscape over a diverse population of sequences but I believe this particular dataset introduces important considerations as a more complex dataset. At the "local" level stability can be affected by cellular/environmental conditions (eg pH), species-specific factors such as glycosylation so I'm not sure this is indicative of a model's ability to learn useful sequence-based representations of similar sequence variants across species/contexts. Stability is also influeced strongly by factors such as size, disordered regions, AA & SS composition and complexing with other proteins which complicates assessment at the "global" sequence level. These sorts of signals might indicate why the pretrained models are so much more successful. I also suspect that solubility may be a confounder of the measurement but accept this would be a limitation of the data collection method rather than the authors' work. Furthermore, measurements are given across a variety of cell lines within and between species (across which measurements can vary) and I believe there are replicates present in the dataset. It's not clear to me how these are addressed and I think more careful splitting and curation strategies might be required such as based on CATH/SCOP classifications or splitting the data into several tasks (eg. species/cell-line specific, human, human-secreted, eukaryotic etc) to account for the different contexts and eliminate non-sequence-based confounders as much as possible. In its current form it's not clear what insight good or bad performance on this task would give me into my model  (or its biological relevance) and its strengths/limitations due to the many externalities. At the very least I strongly believe there should be more discussion of these limitations /nuances as a warning to practitioners & to prevent inadvertent misuse.

## Minor
The authors present well-motivated splitting strategies that are informed by domain knowledge and show empirically that this is a well-motivated problem to be addressed in a meaningful benchmark. I believe the authors can consider additional splitting strategies. For instance, as I'm sure the authors are aware, not all amino acid substitutions are created equal (eg similar sized/charged substitutions are more likely to have similar impacts on fitness). As an example I would propose clustering substitutions by the properties of the introducted amino acid(s) and creating splits based on these clusters as an additional task. This is a suggestion rather than a strict concern & would be interested in hearing the authors' thoughts on this.

Do the authors expect to add tasks to the benchmark as datasets become available. If so, can they indicate which types of additional datasets will be complementary to the tasks they have constructed?


I believe the addition of datasheets would be useful to summarise the datasets.

**Additional Feedback:**

I would like to commend the authors for their interesting work.

**Clarity:**

The paper is well-written and accessible. The introduction motivates the creation of the benchmark and contextualises the problem for a lay ML audience.

**Documentation:**

Documentation of the codebase could be improved. I appreciate the inclusion of notebooks detailing the dataset preparation, though these could be made easier to follow - there is a burden on the reader that is not necessary. The inclusion of datasheets would be useful for users.

**Ethics:**

I beleive the authors address the ethical implications sufficiently.

**Relation To Prior Work:**

Related work is addressed well.

**Summary And Contributions:**

The authors introduce FLIP, a benchmark for protein function prediction motivated by current shortcomings in datasets for protein engineering. FLIP comprises three tasks covering protein thermostability prediction, a mutational screen along an AAV binding interface, and a combinatorial mutational screen on GB1 assessing binding fitness. The authors formulate multiple tasks with the datasets using well-principled splits which are a good contribution to the ML-assisted protein engineering field. The authors train baseline models for the tasks, using a variety of modelling approaches.

---

> ### Author Response · Authors · 2021-09-30
> **Response pt 1**
>
> > Hyperparameter tuning is not performed for baselines. The authors state they expect absolute and relative performance are likely to change following a hyperparameter search which makes it hard justify accepting the empirical results presented. The authors justify this as being a meaningful comparison because "applied users often accept defaults". This seems like a weak justification as a different set of hyperparameters could alter the findings and thus the direction of follow-on research if users were to simply accept defaults. The purpose of benchmarks is to understand the relative merits of different modelling approaches on a relevant problem space and at present the authors do not present this.
>
> As the most consistent point across all you four: loud and clear! We have performed a hyperparameter search as suggested, with the aims of varying regularization and network size when appropriate. We choose one hyperparameter for ridge regression and ESM with 1D attention, and perform a grid search on 3 parameters on the CNN. Existing values used in our first submission are underlined.
>
> Ridge Regression:
> Alpha (regularization) value: 0, 1.0, 2.0
>
> We did not want to vary the solver, so we chose to vary the alpha value in the sklearn package.
>
> ESM with 1D attention:
> Learning rate: 1e-2, 1e-3, 1e-4
>
> We vary the learning rate as a form of regularization here. We do not change the width of the network as the input dimension is dictated by the embedding dimension of ESM, which is 1280.
>
> CNN:
> Kernel size: 3, 5, 7
> [Input, output] size: [256, 512], [512, 1024], [1024, 2048]
> Dropout: 0.0, 0.2, 0.5
>
> We vary the kernel size, the input/output size (keeping the output dimension twice the input), and add regularization using dropout.
>
>
> > Furthermore, I think as a benchmark the authors should provide principled validation splits as this would be highly desirable by users of the benchmark.
>
> Another point shared by reviewer one: please see our response to them.

---

> > ### Author Response · Authors · 2021-09-30
> > **Response pt 2**
> >
> > > I'm not sure I'm convinced that the inclusion of the meltome dataset is especially congruent with the other tasks in its current form. The other tasks focus on mutations along a single starting sequence and effectively sample large portions of the sequence space within a small number of edits. This allows for rich and meaningful characterisation of the fitness landscape. I appreciate the goal of including the meltome task is to increase the diversity of tasks to include a landscape over a diverse population of sequences but I believe this particular dataset introduces important considerations as a more complex dataset. At the "local" level stability can be affected by cellular/environmental conditions (eg pH), species-specific factors such as glycosylation so I'm not sure this is indicative of a model's ability to learn useful sequence-based representations of similar sequence variants across species/contexts. Stability is also influeced strongly by factors such as size, disordered regions, AA & SS composition and complexing with other proteins which complicates assessment at the "global" sequence level. These sorts of signals might indicate why the pretrained models are so much more successful. I also suspect that solubility may be a confounder of the measurement but accept this would be a limitation of the data collection method rather than the authors' work. Furthermore, measurements are given across a variety of cell lines within and between species (across which measurements can vary) and I believe there are replicates present in the dataset. It's not clear to me how these are addressed and I think more careful splitting and curation strategies might be required such as based on CATH/SCOP classifications or splitting the data into several tasks (eg. species/cell-line specific, human, human-secreted, eukaryotic etc) to account for the different contexts and eliminate non-sequence-based confounders as much as possible. In its current form it's not clear what insight good or bad performance on this task would give me into my model (or its biological relevance) and its strengths/limitations due to the many externalities. At the very least I strongly believe there should be more discussion of these limitations /nuances as a warning to practitioners & to prevent inadvertent misuse.
> >
> > Thank you for suggesting great improvements for our work! We do believe that having the meltome in is advantageous, and we do take your points regarding further curation. As such, we are in the process of creating two new splits: the “inter species” (only human), and the “inter cell-line” (only particular cell lines for human). This won’t be finished before rebuttals close, but we ARE working on this and we WILL evaluate it. The CATH/SCOP suggested “redundancy reduction” (also suggested by David Jones in some of his structure prediction work) is also a great idea - due to time constraints: we won’t get there until camera ready. Some of us did this in the past, and whilst not necessarily complicated, it is a very time consuming procedure with many “gotchas”! Wouldn’t want to put in something hastily. We will look into it as one of the next improvements to FLIP.
> >
> > > The authors present well-motivated splitting strategies that are informed by domain knowledge and show empirically that this is a well-motivated problem to be addressed in a meaningful benchmark. I believe the authors can consider additional splitting strategies. For instance, as I'm sure the authors are aware, not all amino acid substitutions are created equal (eg similar sized/charged substitutions are more likely to have similar impacts on fitness). As an example I would propose clustering substitutions by the properties of the introducted amino acid(s) and creating splits based on these clusters as an additional task. This is a suggestion rather than a strict concern & would be interested in hearing the authors' thoughts on this.
> >
> > When designing these tasks we tried to build them around experimental designs that would be applied in a protein engineering laboratory. In this setting, random mutagenesis methods such as site-saturation mutagenesis and error prone PCR dominate, largely because they can often be performed for orders-of-magnitude less expense than more targeted strategies. Gathering data using a random mutagenesis method equates to a random sample from the design space, which is reflected in how splits are “designed” for our benchmarks. In contrast, clustering and splitting by amino acid properties would be a more targeted data collection strategy and, while it would likely result in improved model performance, it may not be the most practical approach from an applications point of view, largely due to the increased cost of gathering targeted data. Overall, we do appreciate the suggestion for a new split, and hopefully this explanation was helpful in explaining why we did not include this one.

---

> > > ### Author Response · Authors · 2021-09-30
> > > **Response pt 3/3**
> > >
> > > > Do the authors expect to add tasks to the benchmark as datasets become available. If so, can they indicate which types of additional datasets will be complementary to the tasks they have constructed?
> > >
> > > Yes, we would not want this effort to die down after submission, but rather as a stepping stone towards a greater initiative. In fact, we added new splits between the first submission and today, and plan to go on adding splits/sets going forward. We hope to leverage FLIP to build a larger community, drawing on experience from organizers of CAFA and other efforts. It would be interesting to add some deep mutational scanning sets (e.g. from DeepSequence), but with an eye on which mutations we believe to truly have the potential to affect function (e.g. at binding residues). This would probably be the “next up”. Furthermore, we feel there’s room for more splits, as you indicated on the meltome landscape, but also on the AAV landscape, as there are more experimental datasets from the Church lab out there with different activity measurements (split = train on one activity, predict the other?). Similarly, there exists another DMS-like dataset for GB1 which could be integrated and compared to what we have already. Also, there could potentially be interesting split ideas that we didn’t consider initially, e.g. one from the community we heard after this went on-line (and which we now added): can we predict high activity by training on low activity (new “few-vs-low” splits in AAV and GB1). Ultimately, the question is not (only) around when datasets become available, but rather which datasets make sense for design, and, especially, which splits make sense! There is one point we strongly considered regarding novel datasets which is using them as private test sets (as CAFA/CASP does): depending on how this effort develops, this would be interesting to us.
> > >
> > > >I believe the addition of datasheets would be useful to summarise the datasets.
> > >
> > > Good idea! We already have READMEs for each split in our repository, but they were quite buried. We will link these better on our website once we have settled on a layout. We also looked closely at the datasheet template and used it to refine our coverage of each dataset in each README.
> > >
> > > > Documentation of the codebase could be improved. I appreciate the inclusion of notebooks detailing the dataset preparation, though these could be made easier to follow - there is a burden on the reader that is not necessary. The inclusion of datasheets would be useful for users.
> > >
> > > Thank you for suggesting we have another pass on the notebooks! We tried to clean them up, add comments, and added a “standard” plot to each split (previously “task”, see comments from reviewer 1), which includes number of samples in train/test and the distributions of the experimental value for each set. We hope this will simplify reading the notebooks! Given time pressure, however, this is a work in progress!

---

> > > > ### Author Response · Authors · 2021-10-07
> > > > **Followup comments**
> > > >
> > > > Dear Reviewer WRMr,
> > > >
> > > > Once again: thank you for your initial reviews. Through our additional hyper-parameter optimization results, revised/added data splits, additional documentation, and more refined commentary we are striving to address your notes. We eagerly await your follow-up comments, should any points remain! We'd greatly appreciate any further comments!

---

> > > > > ### Comment · Reviewer_WRMr · 2021-10-08
> > > > > **Followup**
> > > > >
> > > > > Thanks for such a detailed response. Looks you have a lot planned for FLIP!
> > > > >
> > > > > My concerns around the meltome component of the dataset remain. I appreciate you developing new splits, which is great. However, I’d appreciate if you could expand on your reasoning behind the original formulation being advantageous to include.

---

> > > > > > ### Public Comment · ~Jody_Mou1 · 2021-10-08
> > > > > > **Meltome mixed split comments (1/2)**
> > > > > >
> > > > > > Thank you for your follow-up. We are glad that we could address some of your concerns by curating new splits within the melteome dataset. However, we still view it as important to include the originally-formulated split alongside the newer ones.
> > > > > >
> > > > > > Although the species-specific splits directly address the concern of close protein homologs appearing in both the testing and training splits, we believe that the mixed split still provides a valuable metric for evaluating fundamental aspects of thermostability that transcend species. It's important to note that what codes for thermostability is quite complex and may exhibit varied sequence signatures across families. In particular as shown in Pinney et al, thermostability is a property that can sometimes come down to a single amino acid change, a resolution at which homology-based  models would likely fail. In our work, we aim to enable a sequence-function model that can capture the distribution of sequence patterns that influence thermostability across families. Most/many of these patterns are unknown, may be difficult to explicitly identify or split for, and may range from single substitutions to larger edit distances. We provide the following for reference:
> > > > > >
> > > > > > “Pinney et al. ... reveal the molecular determinants of thermal adaptation in enzymes. For KSI, they observed a trade-off between activity and thermal stability that comes down to a single active-site residue. With their larger dataset, they identified patterns of individual amino acid substitutions that are favored at higher temperatures, and also consider how networks of stabilizing interactions develop.” (https://www.science.org/doi/10.1126/science.aay2784)
> > > > > >
> > > > > > Secondly, the mixed split is also motivated by its relevance to real-world applications in protein engineering, such as random mutagenesis. One example might be directed evolution schemes, which starts with an existing protein and then takes random steps in sequence space, analogous to our approach of taking random samples from the meltome dataset. Work from the Arnold group has shown that stability has been shown to be the most important factor in the evolvability of proteins (https://www.pnas.org/content/103/15/5869), meaning that an accurate predictor of thermostability might lead to meaningful information about the functional landscape of a protein.

---

> > > > > > > ### Public Comment · ~Jody_Mou1 · 2021-10-08
> > > > > > > **Meltome mixed split comments (2/2)**
> > > > > > >
> > > > > > > With regards to species-specific confounding factors that may decrease the predictive power of a broad sequence-function model, this is unfortunately an experimental hurdle present in the original paper by Jarzab et. al. (https://www.nature.com/articles/s41592-020-0801-4#Sec9). However, the authors addressed this point in a number of experiments and analyses, which we highlight and quote below.
> > > > > > >
> > > > > > > Firstly, the authors compared the relative stabilities of orthologous proteins across all 13 species, finding that protein stability is a partly conserved property, which decreases as the sequences diverge:
> > > > > > >
> > > > > > > “Our experiments suggest that protein stability is, in large parts, an intrinsic property of proteins encoded in their primary sequence. But is this property conserved evolutionarily between different species; that is, do proteins maintain their stability as they evolve? To test this hypothesis, we compared relative protein stabilities across all 13 species by computing correlations between the AUCs of orthologous proteins (Methods). Hierarchical clustering of the correlation coefficients (Supplementary Fig. 12) approximately recovered the known phylogenetic relationship between the analyzed species, with two broad clusters for bacteria/archaea and eukaryotes. The eukaryotic subcluster mostly followed the species tree of eukaryotes from yeast to mammals, with the highest correlation observed between mouse and human (r = 0.62), as would be expected by the relatively small evolutionary distance between the two species. These findings suggest that protein stability is at least partly a conserved property but continually decreases as the sequences (and the environmental pressures shaping them) diverge.”
> > > > > > >
> > > > > > > In addition, the authors also investigated the effects of complex cellular environments across species in Figure 3. By comparing proteome profiling experiments within species and within lysates, the authors showed that protein sequence and structure is a dominant determinant of protein stability over cellular environment. This was observed for protein complexes as well:
> > > > > > >
> > > > > > > “It appears that protein complexes are conserved in how tightly they are assembled even though their absolute Tm values are substantially different. We analyzed this more systematically for Homo sapiens, Mus musculus, D. rerio, the EcoCyc database, E. coli, yeast and D. melanogaster, which confirmed that codenaturation of protein complex subunits is often conserved even across distantly related species.”
> > > > > > >
> > > > > > > Lastly, the authors note that there are some factors, such as protein length, that have very little predictive value within one species, but which do correlate across species.
> > > > > > >
> > > > > > > “Therefore, for proteins within one species, the association of Tm and length has little predictive value. In contrast, correlating the mean protein length with the mean Tm values across bacterial species showed a clearer association (Pearson r = −0.59, P = 0.22) and a larger effect (slope of −1.04, Fig. 4b)”
> > > > > > >
> > > > > > > Other potential confounding factors seemed to be poorly correlated with stability, such as protein abundance:
> > > > > > >
> > > > > > > “For E. coli, Leuenberger et al.1 had shown that protein abundance correlated with thermal stability. We observed the same overall trend in most of our data sets, and the effect was more pronounced in nonmelting proteins, a group that the Leuenberger study did not consider (Fig. 4c, mouse bone marrow derived cells, see Supplementary Fig. 8). But again, the correlations are very weak (average Pearson r = 0.16 for Tm and r = 0.17 for AUC) and the effect sizes are very small (average slope of 0.74 for Tm and 0.03 for AUC), so that protein abundance cannot be considered a good predictor of thermal stability.”
> > > > > > >
> > > > > > >
> > > > > > > In conclusion, we view the originally-formulated split as beneficial for modeling purposes to aid protein engineering and decide to keep it within FLIP. With that said, we agree with your original comment:
> > > > > > > “At the very least I strongly believe there should be more discussion of these limitations /nuances as a warning to practitioners & to prevent inadvertent misuse.” We agree with you. We will include and make clear the aforementioned nuances and limitations as it pertains to the melteome dataset in our final manuscript.

---

### Official Review · Reviewer_kx2m · 2021-09-20
**Promising Work in Benchmarks for Protein Function**

**Rating:** 7
**Confidence:** 3

**Strengths:**

This work aggregates data from multiple sources into an easy to use benchmark, which while not the most glamorous of work is indeed useful, necessary and far from trivial since due to the _massive_ space of possible proteins and the limited amount of data that can be collected, generating sound experiments and splits for ML can be quite difficult. ML for protein engineering is becoming a larger field both at ML methods become more sophisticated and the amount of data available grows, so this work definitely has a relevant niche. The authors do a fairly good job at motivating why the tasks they're using are of importance and presenting the data in a mostly clear way.

**Weaknesses:**



The authors do not provide a validation set, which does hurts the accessibility of this work as the authors themselves note determining splits takes some effort to do correctly. The authors note that choosing splits for this work can be difficult, "random splits are notoriously misleading .... as protein sequences are not sampled I.I.D" but they do not provide a validation set for use with each task even they the authors explicitly encourage researchers to do so: "We encourage users to create validation sets from the train sets for hyperparameter optimization". If random sampling from the train set is sufficient to generate a train set due to how the authors generated the splits, this point is moot but it isn't clear to me that this is the case.

Apart from that, there are some specific technical points that I have some worries about, but on the whole I think this work is relevant and has the capacity to be useful for the broader community.

**Additional Feedback:**

Overall I'm super positive about the work! If the authors can address all my points in the clarity category that would be sufficient to move to a 6 in my opinion, although higher would require a bit more discussion with my current understanding.

I have a couple of questions for the authors:
1. What types of mutations are sampled in each of the landscapes? For example, are all the single mutants in GB1 modifications charged AA -> Non-polar AAs? What types of mutations are the ones that are present in the data?
2. A single metric is provided for evaluation, but is this sufficient for a benchmark? This may leads to various failure modes being left unobserved (for example, two models get the same score but they fail to generalize in different ways). Are there any plans to flesh out existing tasks with more metrics/visualizations/probes to better evaluate what failure modes a model may be falling into?
3. Is all the data available? I know the authors mentioned that they downsampled the GB1 data(which may in itself be a problem), but is it possible to download all the raw data so that other groups may try different methods of accounting for the class imbalance? It wasn't entirely clear to me if this was possible from what I could currently see.


## Update
Thank you authors for your response. I'm updating the original review instead of following up due to not being able to get to this until after the discussion period ended. The authors response has been overall satisfactory and I have updated my score to reflect this. More details to come at a later time.

**Clarity:**

In this author's opinion, there are a few points that detract from the overall clarity of paper.

1. The tables with the performance metrics (tables 4-6) do not actually indicate what metric they are measuring. One could reasonably infer that the authors are referring to the spearman correlation coefficient (and this is confirmed by looking at the supplemental tables, where the authors present multiple metrics and hence provide labels), but including this explicitly in the table would help clarity greatly for little added text.
2. While the authors do a good job at motivating why the landscapes included are relevant to protein engineering work, they actually do not really explain what exactly the tasks are. For example, the authors never define "higher-order" interactions for the GB1 landscape, so what exactly is being measured? If the reader has read the previous work or has a background in biochemistry, this wouldn't be too big an issue. However, given the venue, I think the authors should err on the side of overexplainig what exactly the tasks are and what they are trying to measure as this may lead to a better understanding of what exactly the failure mode for each task may look like.
3. Personally, it took me a little while to really understand what was going on in figure 1. I think I get what the authors were going for, a visual representation of the landscapes that the authors propose to evaluate models on and a visual aid for how the splits were determined, but looking at the figure without reading the paper just caused confused me and looking at the figure after reading the paper, figure 1 didn't really add or clarify anything further than what I had already read. Perhaps splitting this figure up into two different figures (one to showcase what the tasks actually are, perhaps actually showcasing the biochemical task beignet solved instead of an abstract task landscape) and another figure to visually represent what exactly the spits were may aid a reader in comprehension.

**Correctness:**

To this reviewers knowledge, all the claims made in this paper are correct but there are a few design decisions that I am a bit skeptical about.

1. This may be a bit on the pedantic side, but the authors I don't know that I agree that each split for a given landscape counts as it's own "task". For each landscape, the end goal is the same and the only difference is how the splits are generated. As such, I don't know that I would agree with calling every individual split it's own "task" without some further elaboration as to how exactly the landscape changes w.r.t the splits that were used.
2. Still on the topic of splits, while the authors did a good job of establishing that random splits may not necessarily be the best idea but:
  a. They randomly sample for GB1 regardless (they do some random sampling in AAV too but with some more constraints)
  b. I don't exactly understand exactly why each proposed split avoids the problems presented by random splits, if the authors could clarify with greater clarity what exactly they are trying to avoid when choosing their splits and how their proposed splitting methods mitigate the those effects that would be helpful.


**Documentation:**

The work is reasonably well documented, although there are a couple of key omission.

1. I was unable to find the scripts used to train the baseline models from the main/supplemental text. Digging around the website a bit was able to lead me to what I presume to be the authors GitHub (https://github.com/J-SNACKKB/FLIP), but I would like to see this more clearly linked to in the text and the authors website: https://benchmark.protein.properties. I have some further comments about this (e.g task naming not quite matching up, etc.) but will refrain from this until the authors clarify if this is the repo they wish to be public facing.
2. The splits provided by the authors don't match the names used in the paper. For example, the GB-1 landscape provide for files labels "four_mutations_task_{1,2,3,4}".csv. I assume this corresponds to 1-vs-rest,..., 3-vs-rest and 4 corresponded to sampled, but ensuring the naming is consistent across the dataset and the paper is fairly important.

**Relation To Prior Work:**

The authors provide an adequate summary of existing work in the space and explain how their approach is different, in it's aims and goals.

**Summary And Contributions:**

The authors present FLIP, a benchmarking tool that aims to evaluate models that take as input a sequence of amino acids and aim to predict how well a protein will accomplish a given function (given only the sequence). The authors process data for 3 different functinoal "landscapes" and aggregate them in FLIP:
* GB1 - asses the effects of high order interactions between mutations
* AAV - Asses the effects of a subset of mutations wrt binding interfaces
* Themorstability - How do mutations affect the stability of a protein across a range of temperatures.

The authors present a benchmark comprised of data for the tasks above and they propose splits to use with the data in addition to training multiple models to serve as baselines.

---

> ### Author Response · Authors · 2021-09-30
> **Response part 1**
>
> > The authors do not provide a validation set, which does hurts the accessibility of this work as the authors themselves note determining splits takes some effort to do correctly. The authors note that choosing splits for this work can be difficult, "random splits are notoriously misleading .... as protein sequences are not sampled I.I.D" but they do not provide a validation set for use with each task even they the authors explicitly encourage researchers to do so: "We encourage users to create validation sets from the train sets for hyperparameter optimization". If random sampling from the train set is sufficient to generate a train set due to how the authors generated the splits, this point is moot but it isn't clear to me that this is the case.
>
> A valid point, which we discussed in great detail prior to submission: how to make validation splits that help yet do not constrain ML practitioners? Emerging model training strategies to increase generalization to challenging “out of domain splits” rely on further segmenting the training data into domains or environments (e..g, ​​ https://arxiv.org/pdf/1907.02893.pdf). This strategy has been further developed with success in molecular prediction tasks (https://people.csail.mit.edu/wengong/assets/RGM.pdf). We believe validation is an important implementation choice and thus is best engineered by each practitioner interested in our landscapes. In practice, existing work standardizing train/test sets (e.g., DeepLoc) does not include validation for precisely this reason .  Thus, we landed on intentionally not forcing practitioners into a fixed validation set to allow for varied training strategies, as would be the case in the real-world protein engineering studies these benchmarks are designed to simulate. However, to your point, we include an extra column in each split with sequences which may be used for validation. This enables practitioners who want an easy benchmark experience to quickly produce test-set numbers. Note that in the text, we highlight that users of the benchmark should consider engineering their own validation splits!.

---

> > ### Author Response · Authors · 2021-09-30
> > **Response pt 2**
> >
> > > This may be a bit on the pedantic side, but the authors I don't know that I agree that each split for a given landscape counts as it's own "task". For each landscape, the end goal is the same and the only difference is how the splits are generated. As such, I don't know that I would agree with calling every individual split it's own "task" without some further elaboration as to how exactly the landscape changes w.r.t the splits that were used
> >
> > Point taken! It’s now “splits” both in manuscript and websites. We may have missed some instances of “task” remaining throughout the text, and will do our best to scrub any that remain.
> >
> > > Still on the topic of splits, while the authors did a good job of establishing that random splits may not necessarily be the best idea but: a. They randomly sample for GB1 regardless (they do some random sampling in AAV too but with some more constraints) b. I don't exactly understand exactly why each proposed split avoids the problems presented by random splits, if the authors could clarify with greater clarity what exactly they are trying to avoid when choosing their splits and how their proposed splitting methods mitigate the those effects that would be helpful.
> >
> > Another great catch! We wanted to analyze how performance increases when using a “straightforward” sampling approach, compared to biologically-informed splits. There are multiple kinds of sampling happening for this dataset. First, there are the biologically motivated splits of “1-vs-rest”, “2-vs-rest” and “3-vs-rest”. After splitting sequences, we use sampling to ensure nonfunctional sequences do not dominate training or test sets. Because this sampling is done after stratification, fundamentally we are still evaluating the relevant type of generalization. This approach has been taken to GB1 in previous work https://www.biorxiv.org/content/10.1101/2020.12.04.408955v1.  On the other hand, we do use a truly random split of sequences that we refer to as “sampled.” This split was intended to highlight the danger of random splitting, not to be used by future users of FLIP. In other words, the “straightforward” sampling splits are “special” in the sense that they rather serve for the purpose of discussion in the manuscript, but can be ignored by practitioners . We have edited the manuscript, moving discussion of random splits to discussion rather than results.
> >
> > > The tables with the performance metrics (tables 4-6) do not actually indicate what metric they are measuring. One could reasonably infer that the authors are referring to the spearman correlation coefficient (and this is confirmed by looking at the supplemental tables, where the authors present multiple metrics and hence provide labels), but including this explicitly in the table would help clarity greatly for little added text.
> >
> > Thank you for pointing this out. We have noted in the captions of tables 4-6 that the metric used is Spearman correlation.
> >
> > > While the authors do a good job at motivating why the landscapes included are relevant to protein engineering work, they actually do not really explain what exactly the tasks are. For example, the authors never define "higher-order" interactions for the GB1 landscape, so what exactly is being measured? If the reader has read the previous work or has a background in biochemistry, this wouldn't be too big an issue. However, given the venue, I think the authors should err on the side of overexplainig what exactly the tasks are and what they are trying to measure as this may lead to a better understanding of what exactly the failure mode for each task may look like.
> >
> > We agree that the term “higher-order” was indeed not defined and upon consideration thought it was best to remove it. For clarity, we instead use more precise language to refer to (1) number of mutations and (2) epistasis separately. We added a brief explanation of epistasis to the text as well as how epistatic effects can constrain evolution and lead to effects that are more difficult to predict. We also rephrased the task as “learning from variants with fewer mutations to predict the activity of variants with more mutations.” Finally, we have added a glossary in the supplement at the suggestion of Reviewer 3 that defines some of the protein engineering/biochemical terms more thoroughly.

---

> > > ### Author Response · Authors · 2021-09-30
> > > **Response pt 3/3**
> > >
> > > > Personally, it took me a little while to really understand what was going on in figure 1. I think I get what the authors were going for, a visual representation of the landscapes that the authors propose to evaluate models on and a visual aid for how the splits were determined, but looking at the figure without reading the paper just caused confused me and looking at the figure after reading the paper, figure 1 didn't really add or clarify anything further than what I had already read. Perhaps splitting this figure up into two different figures (one to showcase what the tasks actually are, perhaps actually showcasing the biochemical task beignet solved instead of an abstract task landscape) and another figure to visually represent what exactly the spits were may aid a reader in comprehension.
> > >
> > > Fair! The point was indeed to have a “graphical abstract” for what we did. We tried to improve the figure to have it “stand on its own two feet” and hope that this will help readers immediately grasp what’s going on (indeed, as most except you four will do: without reading the text). To this end, we added a workflow diagram for use of these datasets and splits, improved labeling and detail within the figure for clarity, and modified the figure caption accordingly.
> > >
> > > > I was unable to find the scripts used to train the baseline models from the main/supplemental text. Digging around the website a bit was able to lead me to what I presume to be the authors GitHub (https://github.com/J-SNACKKB/FLIP), but I would like to see this more clearly linked to in the text and the authors website: https://benchmark.protein.properties. I have some further comments about this (e.g task naming not quite matching up, etc.) but will refrain from this until the authors clarify if this is the repo they wish to be public facing.
> > >
> > > We added items to the website to link users more clearly.
> > >
> > > > The splits provided by the authors don't match the names used in the paper. For example, the GB-1 landscape provide for files labels "four_mutations_task_{1,2,3,4}".csv. I assume this corresponds to 1-vs-rest,..., 3-vs-rest and 4 corresponded to sampled, but ensuring the naming is consistent across the dataset and the paper is fairly important.
> > >
> > > We fixed file names to more closely resemble the ones used in the manuscript. We also fixed the nomenclature of the splits to be more consistent (e.g. few-many in AAV is now “7-vs-rest”, as this is similar to GB1’s “x-vs-rest”). THANKS for the suggestion to standardize even further!
> > >
> > > > What types of mutations are sampled in each of the landscapes? For example, are all the single mutants in GB1 modifications charged AA -> Non-polar AAs? What types of mutations are the ones that are present in the data?
> > >
> > > For the GB1 set, the RAW dataset included mutations for every residue at one of four sites to any other amino acid (a-la deep mutational scan). For this set, we ended up excluding several mutation combinations as the vast majority of probes mutations resulted in non-functional sequences. For AAV, the library of mutations included more variation (e.g., including insertions, deletions of multiple residues, as well as substitutions). For the meltome set, no mutations around a wild-type sequence were made, but rather sequences from different organisms. As some organisms have homologs (similar sequences that diverged in evolution), a naïve observations may be “this mutation changes the thermal stability in this way”, however as the sequences are inserted into widely different “contexts” (organisms), there is no 1-to-1 mapping from sequence change to thermal stability effect.
> > >
> > > > A single metric is provided for evaluation, but is this sufficient for a benchmark? This may leads to various failure modes being left unobserved (for example, two models get the same score but they fail to generalize in different ways). Are there any plans to flesh out existing tasks with more metrics/visualizations/probes to better evaluate what failure modes a model may be falling into?
> > >
> > > Thanks for this observation, shared by a different reviewer! We do provide some more metrics (MSE on the test set, as well as spearman and MSE on the training sets); on the website and in the supplement), and we plan on including further ones with biological relevance in the future.
> > >
> > > > Is all the data available? I know the authors mentioned that they downsampled the GB1 data(which may in itself be a problem), but is it possible to download all the raw data so that other groups may try different methods of accounting for the class imbalance? It wasn't entirely clear to me if this was possible from what I could currently see.
> > >
> > > Yes, we link to where all original datasets come from in our notebooks. We don’t re-distribute the data itself in our repository as GitHub has some pesky size limitations. However, we now added all original datasets stored on an available server (http://data.bioembeddings.com/public/FLIP).

---

> > > > ### Author Response · Authors · 2021-10-07
> > > > **Recognition of support**
> > > >
> > > > > Thank you authors for your response. I'm updating the original review instead of following up due to not being able to get to this until after the discussion period ended. The authors response has been overall satisfactory and I have updated my score to reflect this. More details to come at a later time.
> > > >
> > > > Thank you for the vote of confidence and for the support ~ our work benefitted greatly from the comments by you and fellow reviewers.

---

### Comment · Reviewer_DPa5 · 2021-09-30
**Author Response not yet Visible**

&nbsp;

Just a quick note to double-check that the authors have not yet submitted a response? On occasion, the *Readers* box may be left blank which renders the response invisible to the reviewers.

&nbsp;

---

> ### Author Response · Authors · 2021-09-30
> **Now online!**
>
> Thank you for the heads up! We tried including results until the very last moment, which delayed our response. Apologies! We suggest to go through the responses in the PDF in the supplement (same as below) as they are a bit more "readable".

---

### Author Response · Authors · 2021-09-30
**Thank you for your reviews!**

Dear reviewers: thank you so much for your awesome notes, many of which sparked tremendously interesting discussions with colleagues and amongst ourselves. It would have been great if we could have engaged in a proper “discussion” (maybe: at the conference!), but between running experiments, re-writing notebooks and many other details to fix, 6 days (-2 for those with children) were simply not enough to do it all. The revised manuscript is our best effort with all hands/GPUs on deck. If you indulge us, we will make sure to close all open points before “camera ready”.

To make it easier for you, here are the three main points we tackled:
- Hyperparameter search: this was a common note, and we have now included results from an initial sweep. The relative performance of models has not substantially changed with hyperparameter optimization, leaving our initial conclusions unchanged. We are running a more extensive hyperparameter sweep that will be done by camera-ready.
- Splits (previously called “tasks”): we have normalized nomenclature and approach (e.g. we did 1-vs-rest and 2-vs-rest for AAV). We also added 1 more split idea (low-vs-high) for AAV and GB1 based on community feedback, and we are adding 2 splits to the meltome set based on one of your comments. Results are not yet finalized, but will be ready for camera-ready.
- Clarity of online resources: we are updating them as much as we can. One week has simply not been enough time to get it all right, but we will keep on at it while this review unfolds. Happy if you have any other pointers to what may be unclear!

# P.S.: a pdf point-by-point answer is provided in the supplements for your convenience!

---

> ### Comment · Reviewer_DPa5 · 2021-09-30
> **Many Thanks for the Response and the Amendments**
>
> &nbsp;
>
> Many thanks for the authors for providing a detailed response and making substantial amendments to the paper and codebase! I include a brief response now due to time constraints (before the end of the discussion period!) with a more detailed response to follow.
>
> &nbsp;

---

> > ### Comment · Reviewer_DPa5 · 2021-10-04
> > **Very Satisfied with Author Response; Upgraded Score although Experiencing Issues with Editing the Score on OpenReview**
> >
> > &nbsp;
> >
> > Many thanks once again to the authors for their response!
> >
> > 1. The hyper-parameter sweeps performed in addition to those that will be performed ahead of the camera-ready deadline (given the compute-intensive nature of the models) should hopefully address the defensibility of the claims made in the paper.
> > 2. The included glossary and codebase documentation will no doubt benefit users of the benchmark.
> >
> > **I upgrade my score to 7 although I am having some issues editing the score on OpenReview. The facility would appear to have been withdrawn even though the review period has not yet ended!**

---

> > > ### Author Response · Authors · 2021-10-07
> > > **Thank you!**
> > >
> > > Thank you very much! We are happy to have had the chance to impove our work through your comments!

---

> > > > ### Comment · Reviewer_DPa5 · 2021-10-07
> > > > **Score now Updated**
> > > >
> > > > &nbsp;
> > > >
> > > > Score is now updated in the system!
> > > >
> > > > &nbsp;

---

### Decision · Program_Chairs · 2021-10-09

**Decision:**

Accept

**Comment:**

The reviewers feel that the dataset is useful and complementary to existing benchmarks, but also raised some issues including but not limited to: (1) three reviewers pointed that the hyperparameter tuning was not performed, therefore, some claims may not be convincing and (2) there is no official validation set provided. As a result, the contributions of this work are greatly weakened. The authors' response clarified some important points. In view of that, the authors are strongly invited to take the feedback on board for the final version.